# Work–Care Reconciliation Strategies for a Variety of Informal Carers: What Works and What Does Not?

**DOI:** 10.3390/healthcare13161961

**Published:** 2025-08-11

**Authors:** Tjaša Potočnik, Valentina Hlebec

**Affiliations:** Faculty of Social Sciences, University of Ljubljana, Kardeljeva pl. 5, 1000 Ljubljana, Slovenia; tjasa.potocnik@fdv.uni-lj.si

**Keywords:** reconciliation strategies, working carers, work–care balance

## Abstract

**Objectives**: The aim of this study is to describe and examine reconciliation strategies for reconciling work and informal care that are aimed at different types of working carers (carers of the following: family members with dementia; partners; children under 18; adult children; parents; other family members). **Methods**: We conducted a cross-sectional survey to examine structural strategies for a work–care balance throughout the caregiving period, followed by the frequency of the use of strategies in the last 12 months to better understand what is an effective work–care balance strategy for different working carer types. Hierarchical cluster analysis was conducted in February 2025 on 299 working carers drawn from a representative sample of adult Slovenian residents in an online probability panel. **Results**: The results show five clusters of carers that clearly indicate their use of reconciliation strategies depends on the national context, the work organisation and employees, the family structure, the value orientation of informal carers, and the type of care recipient. **Conclusions**: A variety of policy measures are needed to enable informal carers to remain active in the labour market while they care for dependent family members and relatives.

## 1. Introduction

Changes in society like the growing participation of women in the labour market, population ageing, altered family structures, restructured labour markets, and postponed retirement [1,2,3] have led to a growing number of working carers [1,2,4,5], while the issue of reconciling work and family care is gaining the attention of policymakers and employers [6]. Working carers are individuals who provide (typically) unpaid care and support to someone with a disability, long-term or terminal illness or age-related frailty in a relational, emotional or community context, and at the same time remain active in the labour market [4] (pp. 6–7). Such care is most often provided to ageing parents, disabled or seriously ill children, partners, other family members, friends or neighbours [4,7]. The biggest proportion of working carers is found in the final third of their working life, namely, in the last 10 to 15 years (50–65 years of age) [5]. However, as caring responsibilities can arise at any point in life, it should be recognised that workers of all ages may have informal caring responsibilities and that their reconciliation strategies may vary depending on various individual and contextual factors. For example, childcare responsibilities tend to arise early on in a career (often in one’s twenties or thirties) and continue throughout a career, whereas caring for ageing parents or older family members generally occurs later in a career. Although caring for a partner is most common in older age, it can occur earlier depending on individual circumstances [8,9].

In Europe, informal care continues to reflect deeply entrenched gender norms, with women disproportionately taking on more complex and time-consuming care tasks. Men are increasingly involved in informal care, but usually with less demanding tasks, reinforcing women’s primary role in informal care provision. Gender inequality is less pronounced among those caring for a partner, as male carers are more likely to take on informal caring tasks in partner relationships than in other caring contexts (e.g., caring for parents, parents-in-law, other relatives). Caring for a partner is often unexpected, usually intensive, and can coincide with ongoing childcare responsibilities. If the needs of the care recipient are complex—such as with dementia, which often requires constant monitoring—the emotional and cognitive demands on the caregiver rise significantly [10]. Intensive caregiving, which typically includes personal care (dressing, eating, bathing, etc.), has been found to be negatively associated with employment outcomes [11,12], with clear intersectional inequalities emerging across gender, socioeconomic, and ethnic dimensions. Middle-aged women frequently reduce their working hours, even when the care is on a medium level (10–15 h a week), while intensive care (i.e., care exceeding 20 h a week [7]) is more likely to be associated with exiting the labour market or early retirement [13]. Female caregivers, caregivers from lower socioeconomic groups, and caregivers who live with care recipients are even more exposed to these disadvantages because they are more likely to provide intensive care [14,15,16,17,18].

In countries where formal care services are well developed and accessible (e.g., Nordic countries like Sweden, Finland, and Denmark), research shows the intensity of informal care and issues with reconciling work and informal care are not as acute as in countries where formal care services are unavailable/underdeveloped (such as southern European countries, e.g., Spain, Italy, Portugal) [19,20,21]. Countries that predominantly rely on families to cover the long-term care needs of their significant others and do not provide state support tend to offer limited support for work–care reconciliation, which can disproportionately affect women and caregivers of lower socioeconomic status [16,22]. On the other hand, countries characterised by defamilialism or supported familialism tend to see fewer women withdrawing from the labour market or reducing their working hours so as to provide care since institutional support eases the burden of informal care. This finding was confirmed in recent research by Hlebec et al. [23], who established that working carers in strong care regimes—characterised by accessible, affordable, and high-quality formal care services and supportive labour policies (generous leave, flexible work options, and protection, e.g., Denmark, Germany, Finland, Sweden, France, Belgium, Austria, UK, Ireland, Lithuania)—experience less work–family conflict than working caregivers in weaker care systems where the lack of adequate policies means families must privately organise the care, often at high personal and professional costs [23]. Paid work may be seen as a protective mechanism for working carers as it gives them social and financial security [5,24,25,26]. Nevertheless, when adequate support is missing or the care recipient has intensive care needs, paid work can negatively impact caregivers’ quality of life, physical and mental health, economic stability, and labour market participation [25,26,27,28,29,30].

The strategies working carers use to reconcile paid employment and informal care are shaped by social networks, accessible and affordable formal care services [20,31], day care services (which permit carers to assist care recipients in a supervised setting during working hours) [4,24], financial support, paid care leave, respite care, and the availability of respite care facilities [4,20,24,32,33]. On the organisational level, the availability of flexible work arrangements, a supportive organisational culture, along with the behaviours and practices of managers and employees, have a significant impact on the way working carers manage and reconcile their work and informal care obligations [6,8,24,34].

The previous (mostly qualitative) research has identified a range of reconciliation strategies working used by carers to manage the dual demands of work and informal care. These strategies include reducing or changing working hours, physically separating work and care, working from home, taking care leave, changing careers, taking a career break, changing hours, working remotely, turning down job offers, switching to a family-friendly employer, moving closer to work, becoming self-employed, rejecting promotions, accepting less demanding roles, taking early retirement, or leaving the workforce altogether [16,24,33,35,36]. More drastic measures are taken especially when caregiving is intensive and working carers do not have access to adequate formal and informal support to reconcile work and informal care—such as early retirement or exiting the workforce. The latter is often only the last resort after all other possibilities have been exhausted [33,37]. Supporting working carers is thus crucial for lowering the negative impact of work–care reconciliation on their labour market participation, as is the need to develop adequate support policies that strengthen their resilience and well-being [5,33].

### Contextual Framework

The Slovenian long-term care system has traditionally relied heavily on informal, family-based care with limited state support, an arrangement that reflects the principles of implicit familialism [38,39]. Still, currently the country is undergoing a profound transformation following the adoption of its first long-term care law (ZDOsk) in 2021; namely, a historic step for addressing the needs of the ageing population and its caregivers. After a change in government, an amended version of the law has been in force since 3 August 2023 (ZDOsk-1). Although as a post-socialist country, Slovenia has some similarities with other Central and Eastern European (CEE) countries, it is also distinct from them, especially with regard to the dichotomy between the organisation of care for children and care for older adults. Care for children is defamilialised due to Slovenia’s generous parental leave, the robust family support policies, and the extensive public childcare system [40,41,42]. On the contrary, care for (older) adults remains highly familialised and characterised by limited state support and strong reliance on informal (mostly family) carers who are also legally obliged to contribute to the cost of long-term care (LTC) services if a care receiver’s income is insufficient [39,43]. The availability and accessibility of formal LTC services in Slovenia were the subject of several studies [43,44,45]. Findings of those studies revealed low availability, accessibility, and affordability, as well as large regional inequalities in access to formal care services [43,44]. In Slovenia, the limited accessibility, availability, and adequacy of formal LTC services further limits the set of options working carers have to organise the LTC of their relatives, and thus must often rely on organisational arrangements, informal networks, or market-based solutions to meet their family members’ care needs [39,46]. An important contextual factor influencing the ability of working carers (who are mainly women) to reconcile their paid work and informal care is the enduring high female employment rate in Slovenia (it was 76% in 2020 compared to 71.4% in the EU-27 [47]). Only 8.7% of the total working population was employed part-time in 2020. Among women, the proportion was slightly higher (12.7%), yet well below the EU average (29.2%) for the same period [47]. In Slovenia, part-time work is more common among mothers of young children. This is facilitated by the Parental Protection and Family Benefits Act of 2014, which allows reduced working hours in certain conditions. In contrast, no comparable regulation exists for people who care for (older) adults in need of assistance.

According to the EC, LSE, and Zigante [5] (pp. 20–21), around 15% of the total population in Slovenia are informal carers, whereas working age (18–64 years) people, 13.1%, provide informal care. Among the latter figure, two-thirds (8.3%) are employed and just over one-third are unemployed (4.8%). The biggest gender differences in informal care are seen among carers aged 50–64 among whom 17% of men and 27% of women provide care. In Slovenia, working carers have a right to be absent from work for a shorter period (up to 10 days) and receive compensation for this (amounting to 80% of the person’s average earnings in the previous calendar year). Still, the legislation in this area is narrowly defined. It states that working carers are only entitled to leave and compensation if they provide care to a co-residing family member, such as a spouse or child, but not parents or other relatives (Health Care and Sickness Insurance Act (Health Care and Health Insurance Act, Official Gazette of the Republic of Slovenia, No. 9/1992)). Since 1 January 2024, family carers of older adults in Slovenia have been able to receive a care allowance equal to 1.2 times the minimum wage (pursuant to the mentioned recent Long-Term Care Act (The Long-Term Care Act, Official Gazette of the Republic of Slovenia, No. 84/2023), albeit applying for this benefit requires a caregiver to completely withdraw from the labour market). In addition, 5 days of care leave and the right to work part-time were included in the Employment Relationships Act in 2023 in line with the European Work-Life Balance Directive (EU Directive 2019/1158 [48]). Nonetheless, care leave is unpaid and part-time working arrangements require the employer’s consent, which might limit the uptake and effectiveness of this policy measure.

In a qualitative study on working carers of older adults in Slovenia, Potočnik [46] (unpublished) identified six main strategies used by working carers of older adults to reconcile their work and informal care tasks: adapting the caregiving to work, adapting work to the caregiving (leaving the job to provide care; changes in work organisation; flexible working hours and locations), psychosocial strategies, adjusting employment status, transferring rights and transferring responsibilities (to other family members; outsourcing the caregiving (to formal carers or to a wider social network)). These strategies were included in the questionnaire used for the study discussed in this article. The mentioned strategies were coupled with strategies found in previous research [16,24,33,35,36].

The fact that the use of different strategies to reconcile paid work and informal care is inextricably linked to working carers’ embeddedness in national, organisational, and family contexts along with the availability and accessibility of appropriate support measures [31,33,49] makes it crucial to investigate how strategies vary among different types of working carers. Targeted support measures can be developed on this basis to prevent negative consequences such as early exits from the workforce or the exacerbation of existing inequalities, particularly for women and carers who are already socially or economically disadvantaged [16,33].

## 2. Methods

Data from the Survey on Population Aging and Dementia 2025 were used for the study. The survey was conducted as part of the project Translational Research on Multi-Target Compounds for Treating Neurodegenerative and Mental Disorders, work package Public Opinion Research on attitudes towards neurodegenerative illnesses and perception of care by informal carers of family members with dementia (ARIS, contract no. SN-ZRD/22-27/0510). The target population is the adult population aged 18 and over with usual residence in the territory of Slovenia, entailing a probability online panel with 10,648 participants. Among the panellists, 1942 were invited to participate and a total of 1549 surveys were fully completed, with 28 being partly completed. The response rate relative to the total number of panellists is 15%, while the response rate relative to the invited respondents is 81%.

The online questionnaire was open for data collection between 4 February 2025 and 28 February 2025. The first invitation was sent by email on 31 January 2025 and contained a link and personalised code to access the online survey. The questionnaire was created using the 1KA application (https://www.1ka.si/d/en). All panellists who were invited to participate received a EUR 5 voucher together with the invitation in advance. Three reminder letters were sent to the selected participants, alternately by email and by post (the first to participants with a known email address, the second to participants without an email address); the letters were dated as follows: by email (6 February, 13 February, 20 February), by post (11 February, 20 February). Two weightings were applied as the informal carers formed an oversample to examine the reconciliation strategies of different types of informal carers. The ranking method was conducted using the SurveyWeightingGUI tool (https://enklikanketa.shinyapps.io/SurveyWeightingGUI/, accessed on 1 March 2025.) in R software using the anesrake package (https://rdrr.io/cran/anesrake/src/R/anesrake.R, accessed on 5 March 2025). Two weights were created:A weight for all respondents: The weighting variables were caregiver (survey data from June 2024), gender x age, region, type of settlement, and education. The design effect is 2.6, and the increase in sample variance due to the weighting is 160%.A weight for caregivers only (based on demographic data of caregivers from the June 2024 survey): The weighting variables were gender x age, region, type of settlement, and education. The design effect is 1.3, and the increase in sample variance due to the weighting is 31%.

Depending on the type of analysis, one of two weightings was used: a weighting for all respondents to reduce the impact of the over-selection of caregivers, and a weighting for carers only based on demographic data from the survey of carers conducted in 2024 (see [50] for more details). A total of 166 variables, including demographic variables, were measured. The median survey duration was 14 min and 14 s. The collected units were reviewed for adequacy and data quality according to the following quality criteria.

Basic Quality Criteria:Consistency of birth year: register vs. survey response;Consistency of gender: register vs. survey response;Household income must be equal to or greater than personal income;Inconsistencies in employment responses to questions Q20 and Q21.

Additional Quality Criteria:More than 10 people in the household;Open-ended questions and numeric input questions;Has children or has children of a partner and is younger than 20;Highest personal income class; over 6000;Retired and younger than 50;Holds a Ph.D. and age less than 26.

Units where more than one inconsistency was detected based on the above criteria were excluded from the database. A total of 56 units were excluded. The data set is securely stored on the password-protected computer of principal investigator V.H.

For the part of the analysis presented here, only respondents with a family member in need of care due to dementia or another chronic (long-term) physical or mental illness or disability were considered with respect to work–life balance strategies. All family carers were included, including those who had recently retired due to caring responsibilities, although most respondents were informal carers.

The construct of interest in the first part of the analysis is the reconciliation strategies used by respondents with dependent family members over the course of their caring history to reconcile their work and caregiving responsibilities. This allows us to compare the reconciliation strategies relied on by self-identified informal carers of certain care recipients as opposed to other carers. The items used in this study are a combination of newly developed items and items that have been used in cross-national surveys such as the European Quality of Life Study (items on flexible working hours). The items developed for this study were derived as follows: A Ph.D. student (T.P., a co-author) conducted a large qualitative study to identify and classify the work–care balance measures used by informal carers of older people. Her work entailed conducting a large number of qualitative in-depth interviews, performing thematic analysis of reconciliation strategies, and compiling an extensive list of reconciliation strategies. Her dissertation is forthcoming. T.P. was also a young carer who during her school years cared for her grandmother. The other co-author, V.H., is currently leading a basic national project to investigate discrimination against carers of children and/or older people in the workplace. A large number of in-depth interviews have been conducted and the material is presently in the thematic coding phase. V.H. is a carer of a child with autism and thus brings her life experience of balancing work and care in a public workplace in Slovenia. The classification of reconciliation strategies was further developed into survey items by V.H., who has extensive teaching and research experience in survey design. The reconciliation measures were reviewed based on interview results of the basic project and lived experience, and further categorised as structural changes which might only occur once in a caring career, and other more common reconciliation strategies, which might occur daily, were rated according to the frequency of occurrence in the last 12 months. Before designing the questionnaire, another colleague, T.R., who had cared for her mother with dementia in the past, additionally confirmed that the items were truly reflected her experience. While T.P. is a qualitative researcher, V.H. holds extensive knowledge of statistics and methodology in both teaching and research. She specialises in survey design with a focus on cognitive laboratory techniques for pre-testing questionnaires.

The list of reconciliation strategies is presented in Table 1.

The second construct for work–care balance assessed the last 12 months of caring with a list of options being presented to respondents with family members in need of care to balance their work and caregiving responsibilities (Table 2). This refers to the list of potential reconciliation strategies used in the past 12 months (Please indicate how often you have taken the following actions in the past 12 months: 1—Everyday, 2—Multiple times per week, 3—Multiple times per month, 4—Multiple times per year, 5—Less frequently, 6—Never). The higher the value of an item, the more frequently the strategy is used. It should be noted that most of the reconciliation strategies are universal, and thus we assume they are relevant regardless of the national care context of a post-socialist country like Slovenia.

Family care tasks (care-related items, Table 3, Table 4 and Table 5) were assessed as follows.

First, a series of questions was asked to determine whether a family member had dementia (response category 1—yes, 0—no).

**Table 3 healthcare-13-01961-t003:** Carers of family members with dementia.

Caring Relatives of Dementia Patients	Value
Do you know someone who now has dementia?	0—No, 1—Yes
Did you know someone who had dementia in the past?	
Were any of the people you know (or have known) who have dementia family members?	
Are or were you one of the people most involved in making decisions about the care of a person with dementia or supporting them financially?	
Are or were you one of the people most involved in the day-to-day care of someone you know with dementia?	

Another set of questions assessed the presence of family members with chronic (long-term) physical or mental health problems, illnesses, or disabilities (Table 4).

**Table 4 healthcare-13-01961-t004:** Family members with chronic (long-term) physical or mental health problems, illnesses, or disabilities.

Do You And/or Your Family Members Have a Chronic (Long-Term) Physical or Mental Health Problem, Illness, or Disability? Please Exclude the Person with Dementia Mentioned in the Previous Questions. By Chronic (Long-Term) Illnesses or Health Problems, We Mean Illnesses or Health Problems That Last (Or You Expect to Last) 6 Months or Longer.	Value
You yourself	0—No, 1—Yes
Your partner	
Your children, younger than 18 years	
Your children, older than 18 years	
Parents or grandparents of you or your partner	
Your grandchildren or your partner’s grandchildren	
Other family members	

Another set of questions assessed the frequency of care for family members in need of care (Table 5).

**Table 5 healthcare-13-01961-t005:** Frequency of care services.

In General, How Often Do You Participate in Personal or Practical Activities (e.g., Housework, Helping with Chores or Schoolwork) for the People Listed Below?	Value
Your partner	1—Daily, 2—3–6 times a week, 3—1–2 times a week, 4—More than once a month, 5—Once a month or less, 6—Never
Your children, younger than 18 years	
Your children, older than 18 years	
Your parents or your partner’s parents or grandparents	
Your grandchildren or your partner’s grandchildren	
Other family members	

Another set of questions on frequency assessed the number of hours per week of care spent on family members with care needs (Table 6).

Further questions assessed the characteristics of the respondents (Table 7) and work (Table 8).

The following methods of analysis were used: Hierarchical cluster analysis based on a search for similarities and dissimilarities of reconciliation strategies used by informal carers. Such analysis was chosen because it is an exploratory analysis and the number of clusters is a result of the analysis, as is the composition of the clusters. Squared Euclidean distance was used as the dissimilarity metric with z-standardised variables, followed by the application of Ward’s clustering algorithm. Ward’s method was used and squared Euclidean distance was applied as a dissimilarity measure [51]. A dendrogram was used to determine the number of clusters. Depending on the type of variable, the clusters were further analysed either by contingency tables, calculation of the percentages in the clusters, or ANOVA to determine the mean values in the clusters.

## 3. Results

### 3.1. Structural Reconciliation Measures

The reconciliation strategies used by respondents with family members in need of care during their caring period to reconcile their work and care responsibilities are shown in Table 9. A weighting for family carers was used. The table is structured as follows: The list of strategies is presented in rows, as is the status of the informal carer; the columns indicate the care recipients. The percentage of respondents using a strategy is presented in columns, with *p* ≤ 0.05 indicated by * next to the number of cases in the last column, meaning that Chi^2^ value was statistically significant.

We first interpret the adoption of specific strategies for different types of carers (Table 9). Part-time employment is most commonly adopted by informal carers of children under and over the age of 18, and by informal carers of other family members. Since 17 respondents are informal carers of children under 18, the Chi2 test is not significant for this category, but is significant for informal carers of children over 18 and informal carers of other family members. The search for a new, more suitable position within the current employment organisation is most frequently assumed by informal carers of children under 18 years of age, followed by informal carers of children over 18 years of age, followed by informal carers of people with dementia.

The difference between informal carers and non-caregivers (Table 9) is statistically significant only for informal carers of family members with dementia. Leaving the current job and looking for a new job with better work organisation was most frequently chosen by informal carers of children under 18 years of age, followed by informal carers of other family members, informal carers of children over 18 years of age, and informal carers of parents. Chi2 was statistically significant for all of the above groups of informal carers, except for informal carers of children younger than 18 due to the small sample size. Although leaving the workplace to become a family assistant is not a common strategy, it was most frequently used by informal carers of grandchildren.

Leaving the labour market during informal caring is not a common strategy (Table 9); it was most frequently chosen by informal carers of grandchildren; however, Chi2 was statistically significant for informal carers of family members with dementia. Leaving one’s current employment and becoming self-employed was most frequently chosen by informal carers of children under 18 and informal carers of grandchildren and parents. The latter Chi2 was significant. Retirement as an adjustment strategy of informal carers or respondents with family care needs shows interesting patterns as this is a common strategy of informal carers of grandchildren, children over 18, and partners. The only statistically significant Chi2 was found for informal carers of other family members.

Looking at the columns in Table 9, we see that different types of carers choose different strategies as common options, e.g., for carers of people with dementia, leaving an organisation and seeking employment in a new organisation is the most frequent strategy (27% and 25%, respectively), followed by part-time employment (18%) or retirement (20% and 16%, respectively). For partner carers, the most common option is retirement (34%), followed by part-time employment (23%) and leaving the current organisation and seeking employment in a new organisation (16%). Carers of children (either younger than 18 or older) follow similar strategies, such as working part-time (56% and 39%, respectively), leaving the current organisation and seeking employment in a new organisation (50% and 32%, respectively), changing positions within the same organisation (40% and 25%, respectively) or becoming self-employed (40% and 10%, respectively). Carers of parents or grandparents were most likely to leave the organisation and seek employment in a new organisation (28%) or seek part-time employment (20%). Carers of grandchildren were most likely to change jobs within the same organisation (100%), retire (50%), or become the grandchild’s official carer (33%). Carers of other family members were most likely to seek part-time employment (44%) or leave their current organisation and seek a position in a new organisation (34%).

The variability of the common reconciliation strategies allows us to see that family carers choose different reconciliation strategies, which most likely depend on their life situation such as age, family structure and, of course, the severity of the care needs of the person requiring care. Among the respondents, completely leaving the labour market and taking formal family care leave or becoming unemployed while providing the care were not very common (Table 9).

### 3.2. Day-to-Day Reconciliation Measures

We additionally investigated the frequency of the reconciliation strategies used by the working carers in the previous 12 months. The reconciliation strategies underwent hierarchical cluster analysis to determine whether we could categorise informal carers in several different clusters based on which strategies they had used. The dendrogram (showed we could examine 2–5 cluster solutions. After thoroughly examining the clusters, the solution with five clusters was chosen for presentation. First, the mean values of the reconciliation strategies between the clusters are displayed in Table 10.

In cluster 1 (Table 10), reconciliation strategies were used less frequently (except for adapting the care to work life, outsourcing the care to other family members, and using lunch breaks for the care, e.g., for phone calls). But even these three strategies were very rarely used. It seems the care needs for carers in this cluster are less intense since they are able to reconcile the care and work with only a few strategies.

Cluster 2 (Table 10) is the second-most active cluster in terms of the use of reconciliation strategies. It resembles cluster 5, with a few exceptions. We first consider which strategies appear most frequently in cluster 2. These include flexible working hours, working from home, and taking exceptional days off. Overtime to make time available for caregiving is also quite common, yet slightly less so than in cluster 5. Interestingly, use of sick leave for caregiving purposes is the most common in this cluster, despite use of this strategy having been quite low. Like cluster 5, outsourcing work tasks to colleagues is quite common, along with not taking on certain work tasks due to the caring. Other strategies include rejecting a promotion at work, taking annual leave, and outsourcing care to family members. These carers employed a variety of strategies, which may have depended on the timing and intensity of the carer’s care needs, and appear to have had a fairly supportive and flexible workplace since the organisation was offering a range of work–life balance options, from flexible working hours to working from home and taking overtime or special leave. In addition, carers could share their workload with colleagues by giving up certain work tasks and promotions. These carers also had supportive family structures that could take on excess caregiving duties as required.

Cluster 3 (Table 10) includes carers who would not reconcile their work and care; they rarely used reconciliation strategies. These may refer to carers of children who were entrenched in the education system and used the public school system, which offers a longer coupling of school and after-school care, or perhaps adult children who were participating in further education or training or activities in day centres for people with disabilities. It appears that the need for care did not interfere with working hours to the extent that it called for compensatory measures.

Cluster 4 (Table 10) represents family carers who used only four effective strategies, among which the most frequent is outsourcing care to formal carers. In fact, in this cluster outsourcing care to formal caregivers is the strategy that was the most frequently used compared to all other clusters, followed by adapting the care to the workplace, outsourcing it to other family members, and using lunch breaks to address caregiving issues. Given that this is the only group that outsources formal care, it is possible all other options had been exhausted by the carers, such as those in an inflexible work environment, and in order to keep their jobs they radically adapted the informal care by including formal care arrangements (on average several times a week) and involving other family members in the care.

Cluster 5 (Table 10) includes family carers who often relied on a variety of work–life balance strategies in a flexible work environment by taking advantage of flexible hours, working from home, using overtime, and (most often, several times weekly to several times monthly) working on evenings, mornings, nights, weekends, vacations and public holidays, and annual leave, and also adapting the caregiving to their work demands. They had support staff and could delegate work tasks, gave up a promotion and certain work tasks, used lunch breaks for the care, and outsourced it to other family members. The only two strategies family carers in this cluster did not use are outsourcing to formal care and taking sick leave to perform the care. These carers may have had very frequent caregiving obligations that were also very intensive (the highest number of hours on average), yet they had such a supportive work environment and family structures that allowed them to keep up with the caregiving and work at the same time, prioritising informal care over formal care.

### 3.3. Organisational Support and Structural Changes

In the next step, we examine how supportive the working environment really was and which structural measures the carers had already taken during the care (Table 11).

Cluster 1 (Table 11) is not characterised by particular features of the workplace, which was moderately supportive, and carers did not make any structural changes to the workplace, apart from 25% of them who left the work organisation and found another job. Cluster 2 contains carers with the most supportive workplace structure, where all measures are most common. A smaller share (17%) created their own supportive working environment by becoming self-employed. In cluster 3, no characteristic stands out. The work organisation was supportive, and the cluster has the lowest percentage (19%) of family carers who left their work organisation due to the caregiving and found a job in another work organisation. However, cluster 4 evidently represents inflexible work environments or professions since only 53% of them provided flexible working hours and 67% special leave. Overtime and the ability to use it for care are readily available (67%). Cluster 4 features the highest percentage of family carers who made structural changes in the workplace, such as working fewer hours (44%), leaving one job for another within the same organisation (56%), leaving the organisation company and finding a job in another one (53%), the biggest share of both those who were unemployed during the caring period (6%) and those (28%) who became self-employed. Cluster 5 represents a moderately flexible workplace, one providing flexible working hours (75%) and the possibility to use overtime to make time available for care (70%), but with the smallest share of those offering special leave (50%). Cluster 5 contains family carers who made structural changes in the workplace, but not as much as in cluster 4. About 39% found a more suitable job within the same organisation, 33% changed work organisation and found a better work organisation with respect to caregiving, and around 17% had recently retired.

### 3.4. Composition of the Clusters

Next, we look at the proportion of different types of carers in each cluster (Table 12).

In cluster 1 (Table 12), we most frequently find carers of parents (82%) and other relatives (79%), followed by carers of children under 18 (75%). In cluster 2, we find carers of partners (100%), children under 18 (100%), followed by carers of parents (75%). In cluster 3, we most frequently find carers of children under 18 (100%), followed by carers of parents (78%). Cluster 4 contains carers of all types, except carers of partners and children over 18. In cluster 5, there are carers of children over 18 (89%), and then carers of parents (74%). The distribution of the different family carer types makes it clear that strategies for reconciling work and care depend on who is being cared for. However, this is not the only context that influences the choice of reconciliation strategies. We next turn to the frequency and intensity of the care-related obligations in terms of hours spent to that end (Table 13).

We first examine the frequency and intensity of care provided in the individual clusters (Table 13). In cluster 1, children under the age of 18 (2.9) and parents or grandparents (2.9) were most frequently cared for, with these two groups on average having received care 1–2 times a week. The intensity of care is low and ranges from an average of 5.4 h (partner) to 13.8 h (child over 18) per week. Cluster 2 contains frequent carers of partners (1.5) and children under 18 (1.4) who provided care between once a day and 3–4 times per week. Caregiving is quite intensive, with partners having provided 75.7 h of care per week and children under the age of 18 receiving an average of 44.7 h of care per week, followed by 32.8 h per week for other family members. Cluster 3 contains several family carers of children under the age of 18 (1.2), who on average cared every day, with a high intensity of care of 55.8 h per week on average.

Cluster 4 (Table 13) includes frequent carers with low care intensity across several categories of carers. Carers of children under 18 (1.1) and carers of children over 18 (1.0) cared on average every day, as well as frequent carers of parents and grandparents (2.7) and other family members (2.9) who on average provided care 1–2 times per week. The intensity of care is low, as is the average number of hours providing care, and the highest intensity is seen among carers of children under 18 (22.2). Cluster 5 contains carers with both a high care frequency and a high care intensity. In cluster 5, we find carers of children under and over 18 who cared every day (1.0), followed by carers of partners who cared 3–4 times a week on average (2.3). While care intensity for the partner is low (8.7 h per week), care for children under 18 is overwhelmingly high (60 h per week on average), followed by high care intensity for parents and grandparents (22.1) and other family members (22.5) hours per week on average. Since these two groups were cared for less frequently, one may assume that the high care intensity for parents and grandparents and other relatives is shared within the family. Even though the frequency of care for children over 18 is lower when looking at the different carer types, the intensity of care is still high (11.8 h per week on average) for all carers of children over 18. This section of the analysis provides insights into the way caring responsibilities influence the choice of reconciliation strategies that working carers tend to adopt. It is clear that both the frequency of caregiving and the intensity of caregiving impact the choice of work–care reconciliation policies among working carers. We next investigate the other characteristics of work in Table 14.

### 3.5. Characteristics of the Workplace

With regard to average weekly working hours (Table 14), cluster 4 reveals the lowest average weekly working hours. This is not surprising because the cluster contains working carers who took on part-time employment to help them reconcile their work and caring responsibilities. In the other clusters, the values are quite similar, ranging from 37 to 43 h per week. Commuting time to/from work is quite similar in all clusters, except for cluster 4 where most part-time employees and self-employed are located. The probability of losing one’s job is not very high, on average “not likely”. As was expected, the proportion of self-employed family carers is highest in cluster 4 (27.8%), while the proportion of employed family carers in the public sector (16.7%) is also lowest in cluster 4. The ease with which one can get by on the household income is quite similar in all the clusters, suggesting the reconciliation strategies of working carers in Slovenia do not depend strongly on income. However, in cluster 4, where outsourcing to formal care is common, the family income score is the lowest. The lowest value is seen in cluster 4 (3.1) and the highest in cluster 3 (4.0). Finally, we turn to the family structure and demographic characteristics of the respondents.

### 3.6. Family and Demographic Characteristics of Carers Within the Clusters

The family and demographic characteristics are shown in Table 15.

The proportion of women is smallest in cluster 2 (37.1%) (Table 15), which is not surprising since the partners’ carers are frequently men. The highest proportion of women is found in cluster 4 (55.6%) where children under the age of 18 and family members with dementia as well as parents and other relatives are cared for. As far as social class positioning is concerned, carers in cluster 4 score the lowest on average (4.3) and those in cluster 3 the highest (6.0). Given the structure of carers in cluster 4, this is hardly surprising (the biggest share of part-time employees and self-employed). Political and value orientation from left to right reveals that cluster 5 is more to the right (6.2) and cluster 3 more to the left. Again, this is expected given the high frequency and intensity of care in cluster 5. With respect to the respondents’ subjective assessment of their own health, the results do not vary much, i.e., on average, subjective health is rated the lowest in cluster 5 and the highest in cluster 3.

In view of the care burden in cluster 5, this is not surprising (Table 15). The age of respondents differs somewhat among the clusters, with the youngest respondents in cluster 3 (38.5) and the oldest in cluster 5 (48.5). In terms of urbanisation, no major differences are evident between the places of residence, with the most rural found in cluster 1 (3.5) and the least rural in cluster 3 (2.9) and cluster 4. The frequency of church attendance is lowest in cluster 4 (7.9) and highest in clusters 2 and 5 (6.0). In terms of household size, there are no major differences among the clusters, ranging from (3.1) in cluster 4 to 4.0 in cluster 2. The percentage of households with children extends from 47% in cluster 3 to 88% in cluster 4. The percentage of partners ranges from 61% in cluster 4 to 82% in cluster 2, which is expected because cluster 2 has a high proportion of partner carers. The percentage of grandchildren living in the household is relatively low, yet the highest in cluster 1 (5%). Living with one’s parents or grandparents is quite common in clusters 1 (42%), 2 (47%), and 5 (42%) and rare in cluster 4 (6%). The percentage of other relatives living in the household is relatively low and the highest in cluster 3 (23%).

## 4. Discussion

The results of this study suggest that reconciliation strategies might not always be a matter of individual choice of working carers, and may instead be shaped by the influence of individual and contextual factors on the strategies they are able to use to reconcile their work and care obligations. The analysis revealed that reliance on certain strategies—such as the use of leave restricted to household members—is shaped by national legislation and policy frameworks, while others are influenced by the work environment, family structure, the nature of the caregiving relationship (including the care recipient, and the frequency and intensity of care), and the value orientations held by working carers. Based on these factors, working carers can be placed in different clusters that differ according to the type of reconciliation strategies they employ. In the table below (Table 15), the characteristics of the clusters are broadly outlined to provide a deeper understanding of how the individual characteristics of working carers and their contextual embeddedness on the micro, mezzo, and macro levels influence their use of different balancing strategies and structural changes to reconcile their work and informal care obligations.

Although previous studies found that intensive caregiving is related to changes in labour market participation, notably among women [7,13,16], our study shows the relationship is somewhat more complex as the impact of the intensity of caregiving obligations on the working carers’ labour market participation appears to be significantly mitigated when organisational and public support is available. When caregiving is not intensive or publicly available support lowers the direct burden of care on working carers, supportive work environments (clusters 1, 2, 3) reduce the need for structural changes in labour market participation (such as working fewer hours, changing jobs, retiring or leaving the labour force altogether).

The results (Table 16) permit us to conclude that cluster 1 represents working carers at the beginning of their caregiving journey because the fact the care they provide is of low frequency and intensity means they might be able to manage to reconcile their work and caregiving responsibilities by using only a few balancing strategies and structural changes in their labour market participation. This might mean that working carers in this cluster can provide care outside of their working hours (e.g., in the afternoon) and/or draw on the support of other family members as they live in multi-generational households. However, if the intensity and frequency of caregiving in this cluster increases, working carers’ may need to adopt more balancing strategies and/or make structural changes in their labour market participation, leading them to move into cluster 2 or 3 depending on their individual and contextual characteristics. Similarly to cluster 1, cluster 2 contains working carers who are employed in a supportive environment, use a variety of work–life balance strategies, and have made some structural changes in their labour market participation. Still, unlike working carers in cluster 1, carers in cluster 2 provide high-frequency care and intensive care for partners and children under 18 and other family members. The predominance of men in this cluster is related to the high prevalence and intensity of partner care in it. The fact that most working carers in this group are highly educated men with high socio-economic status could also be due to existing gender inequalities in the labour market, better work–life balance opportunities in higher status occupations, and the availability of family support since they live in multi-generational households.

Of particular interest is cluster 4 (Table 16), which includes working carers employed in a rigid organisational environment offering limited workplace flexibility. In this cluster, working carers have made almost all structural adjustments possible (e.g., part-time work, self-employment) and, notwithstanding their economic constraints, largely outsource care, often using formal services. The latter is particularly interesting as it points to the fact that formal care services are used by lower-income working carers, which is especially noteworthy as it challenges previous research findings that formal care in Slovenia is mostly inaccessible to lower-income groups. Instead, it indicates a more nuanced reality in which the affordability and accessibility of formal care can vary in different contexts and care situations [43,46]. Alternatively, we could assume that the costs of formal care contribute to financial hardship.

However, as the intensity and frequency of care recipients’ needs grow, employed carers might move from clusters 1 and 2 to cluster 5, where they rely heavily on balancing strategies and make several structural changes to their labour market participation (including by retiring) (Table 16). The limited outsourcing of care in cluster 5 (despite the high caregiving burden) may reflect traditional values that stress the importance of family caregiving in Slovenia. Alongside traditional values, their reluctance or inability to seek support from formal carers might also be shaped by the inaccessibility and inadequacy of formal care services for meeting the care needs of older care recipients in Slovenia, as recent studies suggest [39,45,46]. Nevertheless, cluster 5 shows that even in a supportive work environment very frequent and very intensive care can exceed the limits of workplace support. The fact that working carers in cluster 5 relied on both balancing strategies and structural changes in labour market participation and still encountered difficulties meeting their care obligations suggests the existing institutional and organisational support mechanisms in Slovenia are not sufficient to enable working carers of (older) adults (parents, children over 18, other relatives) to provide informal care without causing negative effects for their labour market participation and well-being [39,46] since they are not accompanied by national policies aimed at mitigating such effects. This is confirmed by the presence of carers of children under 18 in cluster 3, which confirms that strong state support combined with organisational support can help carers to reconcile their paid work and caring responsibilities—even in the context of frequent and very intensive informal care. The availability of publicly provided support is also crucial for shaping working carers’ ability to reconcile work and informal care, particularly by ensuring the availability, accessibility, and adequacy of formal care services for different care recipient types (see, e.g., [20]). This is especially evident with cluster 3, which shows that, in addition to a supportive work environment, access to institutional arrangements reduces the need for work-related adjustments given that working carers in this cluster rarely used balancing strategies and made limited structural changes to their labour market participation. Although the intensity and frequency of care in this cluster is high for children under the age of 18, the direct care demands on parents are low since children with long-term care needs in Slovenia are typically placed in public day care facilities (schools), which often also offer extracurricular programmes for children. A combination of public support and a supportive work environment allows the working carers in cluster 3 to leave their children in a supervised environment during working hours [4,24], reducing the need for balancing and structural strategies. Furthermore, this might reflect the defamilialised orientation of the childcare system in Slovenia where parents typically benefit from a care model that shifts responsibility from the family to the state (defamialisation through state-funded and universally available services) [41,42].

On the contrary, care for (older) adults in Slovenia is strongly normative. Caring for older family members is generally seen as a moral obligation embedded in family life, rather than a social task in its own right. As a result, many carers do not label themselves as carers and are therefore less inclined to seek support in the workplace or through national policies. This lack of self-awareness is exacerbated by the fact that employers and HR managers know little about the needs and rights of working carers. Although many Slovenian companies have family-friendly certificates, these tend to focus on childcare and rarely include policy measures targeting employees providing care for older people. This shows that there is an urgent need to expand and adapt work–care reconciliation measures at an organisational level to enable working carers of older adults to reconcile their dual role in a sustainable way, thus preventing structural changes in their labour market participation and negative consequences for their well-being. In line with previous research [39,42,43], our study highlights the unsustainability of the familialist care system and emphasises the need to create a robust LTC system capable of providing quality care to older adults, thereby reducing the disproportionate burden currently placed on families.

However, the formulation and implementation of policy proposals related to work– care reconciliation and support for working carers requires careful consideration of the specific social, cultural, and institutional contexts. While certain measures can be generalised across the EU (e.g., the Work-Life Balance Directive (EU 2019/1158 [48]), which sets a minimum standard of five days’ leave for carers and the right to flexible working arrangements), the success and appropriateness of specific measures vary considerably from country to country. Nevertheless, recent policy developments in Slovenia, in particular the implementation of the Work-Life Balance Directive (EU 2019/1158 [48]) into the Employment Relations Act and the adoption of the Long-Term Care Act in 2023, hold significant potential to reduce existing gaps in work–care reconciliation policies by improving the recognition and support of working informal carers, not only at the individual and organisational level, but also through coordinated support at the national level.

To ensure these measures are truly effective, they must be adaptable and sensitive to the diverse realities of caregiving. That means recognising different care arrangements, family structures, and employment situations and giving working carers the flexibility they need to navigate both responsibilities. For example, family carers in cluster 1, who are likely to be in the early stages of caring, would benefit from early access to flexible working arrangements and awareness-raising campaigns informing them of their rights and entitlements before care demands increase. In contrast, cluster 4, which consists predominantly of women with low socio-economic status who are employed in less supportive organisational structures, often in the private sector or in precarious forms of employment such as self-employment, points to the need for more robust structural reforms. This could include the introduction of nationally mandated minimum standards for workplace flexibility across all sectors, aimed at reducing the need for carers to make structural adjustments to their labour market participation (e.g., in the form of part-time work or self-employment). Cluster 5 illustrates that even working family carers in supportive organisational contexts can face negative consequences when caring responsibilities are intensive. For this group, targeted psychosocial services, burnout prevention programmes, and the extension of paid care leave would provide much-needed relief and ensure the sustainability of their care arrangements. Here, the French model of converting leave into reduced working hours or the use of psychosocial support and respite services, as is common in Scandinavian systems, could serve as useful policy templates. Additionally, a comparative look at other European countries reveals a wide range of supportive measures—such as generous leave policies in the Nordic countries (see [4,52,53]), flexible part-time leave options in Germany and Austria, and innovative models such as the above-mentioned flexible use of carers’ leave in France to reduce working hours—from which Slovenia could draw inspiration. However, for such measures to be truly effective in the Slovenian context, they need to be adapted and responsive to the country’s cultural specificities, care situation, family dynamics, employment relationships, and the LTC system. Only through such flexibility can the reconciliation of paid work and care become a sustainable and equitable reality for working carers—a reality that supports individual carers, promotes labour market participation and gender equality, and ensures the well-being of both those who provide care and those who receive it.

## 5. Limitations

One of the main limitations of this study is the use of a cross-sectional survey design, which means we can only hypothesise about changes over time. A longitudinal study would certainly be a way to monitor and understand changes over time by selecting new or already used work–care reconciliation measures.

Since we employed individual items and did not construct scales, no further statistical validation was carried out, only the frequency distributions of the reconciliation strategies were checked and attention was paid to missing values. It makes sense to continue developing the measurement instrument and its validation as part of a more structural statistical approach is a suggestion for further work and for future use in other national contexts.

Clearly one limitation is the fact the distribution of clusters could be influenced by the specific national context, i.e., a country that is currently undergoing a major transition from the scattered and uncoordinated provision of services to a long-term care law. Many countries take a more structured approach to long-term care and provide better recognition of informal carers. In any case, while these reconciliation strategies might also emerge in other national contexts, they most likely will be used differently.

While our study explored work–care reconciliation strategies among working carers, it did not address their emotional impact or how these strategies are perceived by the carers themselves. This represents a significant direction for future research, particularly in examining how such strategies relate to perceived caregiver burden and indicators like the work–life reconciliation index. Recent research shows that informal caregiving often imposes substantial physical and psychological strain; however, protective factors—such as resilience, high self-esteem, and a strong sense of coherence—play a crucial role in maintaining caregiver well-being and adaptability to changing care demands [54,55]. Although findings on intervention effectiveness remain mixed, multicomponent and tailored approaches appear most promising for mitigating caregiver burden [56]. Validated tools such as the Zarit Burden Scale or other tools (see, for example, [57]) could be employed in future studies to assess how different reconciliation strategies affect subjective burden. Furthermore, there is a growing consensus that interventions for carers should go beyond alleviating caregiver burden and actively promote carers’ resilience and well-being by incorporating more holistic frameworks [58] and recognising the diversity of caring contexts—an aspect that is often overlooked [56]. To build on existing knowledge, future longitudinal research is needed to investigate how reconciliation strategies evolve and influence carers’ well-being and work–life balance over time.

## 6. Conclusions

Our study highlights the different balancing and structural strategies used by working carers in Slovenia to reconcile their dual role, shaped by a complex interplay of individual characteristics (e.g., gender and socioeconomic status), organisational contexts, care arrangements, and family structure. Our findings suggest that structural changes in labour market participation (e.g., job changes, early retirement or complete withdrawal from the labour market) are more common among working carers employed in less supportive organisational contexts and among those who mainly care for (older) adults. These structural changes are much less common among parents of young children with disabilities, who are more likely to benefit from established reconciliation measures at the national and organisational level. This emphasises the inadequacy of one-size-fits-all approaches to address the different realities of working carers. Additionally, our results show that strategies for reconciling work and care obligations are also shaped by the characteristics of the person in need of care. While frequency and intensity of caregiving remain important factors, the key differentiating factor seems to be the degree to which working carers (depending on whom they care for) have access to adequate support in balancing their dual responsibilities. The uneven development and accessibility of support services across different caring contexts (e.g., childcare versus care for older adults) leads to systemic inequalities in the ability of different groups of carers to maintain their employment, fulfil caring responsibilities, and maintain their general well-being.

In summary, our clusters show that the ability of working carers in Slovenia to reconcile work and care responsibilities is determined by the interplay of individual, organisational, and structural factors. Taken together, these cluster-specific findings emphasise the need for a comprehensive, multi-level policy framework that combines universal reconciliation rights with targeted, context-sensitive interventions tailored to the heterogeneous circumstances of working carers, their employment, and their caring situation. Policies must be gender-sensitive and sustainable and address the different situations of working carers to ensure equal access to support for all. Without such an approach, existing inequalities in the ability of working carers to reconcile work and caring responsibilities are likely to persist or even worsen, undermining gender equality, sustainable labour market participation, and the overall well-being of working carers.

## Figures and Tables

**Table 1 healthcare-13-01961-t001:** The list of reconciliation strategies used throughout the history of care.

Did You Take Advantage of Any of the Opportunities Listed During Your Care Period?	Value
I took on a part-time job.	0—No, 1—Yes
I left my previous job and found a more suitable position in my work organisation.	
I left my work organisation and found a more suitable position in another work organisation.	
I left the labour market and found a job as an official carer for a family member.	
I left the labour market and was not employed during the provision of care services.	
I left my work organisation and became self-employed.	
I left my job and retired.	

**Table 2 healthcare-13-01961-t002:** The list of reconciliation strategies used in the last 12 months.

Please Indicate How Often You Have Done the Following in the Last 12 Months Because You Are an Informal Carer:	Value
Due to caring responsibilities, I took advantage of the right to a flexible start and end to the working day.	1—Everyday, 2—Multiple times a week, 3—Multiple times a month, 4—Multiple times a year, 5—Rarely, 6—Never
I worked from home to provide care.	
I worked overtime due to caring responsibilities to use the extra hours to provide care.	
I used my right to special leave to provide care.	
I worked late evenings, nights, or early mornings to provide care.	
I worked on weekends and/or on public vacations or on holiday to provide care.	
I asked my colleagues to take over some of my work duties because of the caring responsibilities.	
I gave up certain work tasks because of the caring responsibilities.	
I turned down a promotion because of the caring responsibilities.	
I used part of my annual leave for supervision because of the supervision tasks.	
Due to the caring responsibilities, I used my right to sick leave to provide care.	
I adapted my caring responsibilities to my work schedule.	
I hired professional caregivers so I could focus on my work responsibilities.	
I asked other family members to provide care so I could focus on my work.	
I provided care during my lunch break (e.g., made a phone call or delivered lunch).	

**Table 6 healthcare-13-01961-t006:** Frequency of care provision.

On Average, How Many Hours per Week Do You Spend on One of the Following Activities Outside of Paid Work?	Value
Your partner	No. of hours per week
Your children, younger than 18 years	
Your children, older than 18 years	
Your parents or your partner’s parents or grandparents	
Your grandchildren or your partner’s grandchildren	
Other family members	

**Table 7 healthcare-13-01961-t007:** Independent variables.

Variable	Value
Gender	0—Female, 1—Male
Age	Years
Social position	0—The lowest, …, 10—The highest
Political position	0—left, …, 10—right
Attending religious services	1—Several times a week or more, 2—Once a week, 3—2–3 times a month, 4—At least once a month, 5—Several times a year, 6—Once a year, 6—Less often
Household size	Number
Living with children	0—No, 1—Yes
No. of children living in the household	Number
Living with a partner	0—No, 1—Yes
Living with grandchildren	0—No, 1—Yes
Living with yours or your partner’s parents or grandparents	0—No, 1—Yes
Living with other family members	0—No, 1—Yes
Health	How is your general state of health? (4—Very good, 3—Good, 2—Fair, 1—Poor)
Urbanisation (subjective)	1—Big city, 2—Suburb, 3—Small town, 4—Village, 5—Remote farmhouses

… denotes numbers between 1 and 10.

**Table 8 healthcare-13-01961-t008:** Work-related items.

Variable	Value
Sector	0—Not public 1—Public
Self-employed	0—No, 1—Yes
Losing one’s job	Probability of losing your job in the next 6 months (1—Very unlikely, 2—Rather unlikely, 3—Neither unlikely nor likely, 4—Rather likely, 5—Very likely)
Hours spent at main job	In hours
Minutes commuting	Minutes spent commuting
Making ends meet	Making ends meet (1—Very easy, 2—Easy, 3—Fairly easy, 4—With some difficulty, 5—With difficulty, 6—With great difficulty)

**Table 9 healthcare-13-01961-t009:** Structural reconciliation strategies of respondents employed during caring at any given time.

Structural Reconciliation Strategy	NC/IC	Dementia Fin	Dementia All	Partner	Children, Younger Than 18 yrs	Children, Older Than 18 yrs	Your or Your Partner’s Parents or Grandparents	Your or Your Partner’s Grandchildren	Other Family Members
I took a part-time job.	NIC	17	17	**19**	0	0	14	0	11
	IC	18	18	23	**56**	39	**20**	17	**44**
	Num	298	299	97	17	49 *	250	12	140 *
I gave up my previous job and found a more suitable position in my work organisation.	NIC	8	7	17	0	9	8	**100**	18
	IC	15	15	3	40	25	12	**100**	13
	Num	298	298 *	97	16	50	250	11	139
I left my work organisation and found a more suitable job in another work organisation.	NIC	23	**26**	22	0	9	6	0	15
	IC	**27**	25	16	**50**	32	28	17	34
	Num	299	299	97	17	50 *	250 *	11	140 *
I left the labour market and found a job as a formal carer for a family member.	NIC	**4**	2	0	0	0	2	0	2
	IC	3	4	2	0	7	2	**33**	2
	Num	298	299	96	16	49	249	12	140
I left the labour market and was not employed during the provision of care services.	NIC	5	3	0	0	0	0	0	4
	IC	7	9	7	6	3	7	**17**	5
	Num	296	296 *	92	17	47	250	12	140
I left my work organisation and became self-employed.	NIC	8	12	**15**	0	0	2	0	7
	IC	12	9	3	**40**	10	13	17	10
	Num	297	297	95 *	16	50	250 *	12	140
I left my job and retired.	NIC	13	16	31	0	**46**	16	**80**	24
	IC	20	16	**34**	6	31	13	**50**	8
	Num	299	299	97	17	51	249	11	139 *

*—Statistically significant at *p* < 0.05. Dementia fin—decision about care or financial support. Dementia all—most involved in daily care. C—informal carer, NIC—non-carers in the presented category, Num—number of cases, the highest frequency in a row (across all types of carers) as well as the highest frequency in the overall reconciliation strategy is shown in bold. Please note there is an overlap between carers of people with dementia and carers of parents or grandparents, and sometimes they are also included in the category carers of partners.

**Table 10 healthcare-13-01961-t010:** Mean values of reconciliation strategies used in the last 12 months in the different clusters.

Please Indicate How Often You Have Done the Following in the Last 12 Months Due to Your Caring Responsibilities:	CL1 N86	CL2 N36	CL3 N99	CL4 N18	CL5 N36
I took advantage of the right to a flexible start and end to the working day due to the caring responsibilities.	5.79	4.05	5.94	5.18	4.28
I worked from home to provide the care.	5.97	4.45	5.98	5.85	4.93
I worked overtime due to the caring responsibilities in order to use the extra hours to provide the care.	5.81	4.61	5.99	5.93	4.49
I used my right to special leave to provide the care.	5.90	5.06	5.97	5.80	5.58
I worked late evenings, nights, or early mornings to provide the care.	5.69	5.05	5.98	5.77	2.61
I worked on weekends and/or on public vacations or on holiday to provide care.	5.67	4.66	6.00	5.21	3.36
I asked my colleagues to take over some of my work duties because of the caring responsibilities.	5.90	5.11	5.98	5.99	5.09
I gave up certain work tasks because of caring responsibilities.	5.85	5.17	5.99	5.90	5.13
I turned down a promotion because of the caring responsibilities.	5.87	5.56	6.00	5.99	4.83
I used part of my annual leave to provide the care.	5.22	4.70	5.92	5.51	3.75
I used my right to sick leave to take on caring responsibilities because of caring responsibilities.	5.57	5.26	5.97	5.87	5.74
I adapted the caring duties to my work schedule.	4.38	4.41	5.94	3.59	2.39
I hired paid caregivers so I could focus on my work responsibilities.	5.85	5.43	6.00	2.29	5.43
I asked other family members to provide the care so I could focus on my work.	4.85	4.38	5.88	4.18	3.10
I provided the care during my lunch break (e.g., made a phone call or delivered lunch).	4.72	4.54	5.97	3.77	2.62

**Table 11 healthcare-13-01961-t011:** Organisational support and structural changes in the care history.

	CL1 N86	CL2 N36	CL3 N99	CL4 N18	CL5 N36
I had the option of a flexible start and end time, at least to some extent.	62.4%	77.8%	67.3%	52.6%	75.0%
I can use overtime hours to leave early or take a day off.	68.2%	88.9%	71.1%	83.3%	70.3%
If necessary, I can get special leave.	77.9%	88.9%	81.6%	66.7%	50.0%
I have a part-time job.	20.2%	17.1%	17.5%	44.4%	13.5%
I left my previous job and found a more suitable position in my current work organisation.	3.6%	8.3%	17.5%	55.6%	38.9%
I left my work organisation and found a more suitable position in another work organisation.	25.0%	22.9%	19.6%	52.6%	33.3%
I left the labour market and found a job as a carer for a family member.	1.2%	2.9%	1.0%	0.0%	2.8%
I left the labour market and was not employed during the provision of care services.	1.2%	2.9%	3.1%	5.6%	2.7%
I left my work organisation and became self-employed.	3.6%	14.3%	8.4%	27.8%	8.3%
I left my job and retired.	0.0%	0.0%	3.2%	0.0%	16.7%

**Table 12 healthcare-13-01961-t012:** Proportion of different types of family carers in the clusters.

	CL1 N86	CL2 N36	CL3 N99	CL4 N18	CL5 N36
Dementia—decision-making about care or financial support	59.6%	68.2%	47.2%	100.0%	52.4%
Dementia—most involved in daily care	53.8%	68.2%	62.5%	88.2%	57.1%
Your partner	61.5%	100.0%	66.7%	0.0%	n.a.
Your children, younger than 18 years	75.0%	100.0%	100.0%	100.0%	n.a.
Your children, older than 18 years	25.0%	n.a.	66.7%	n.a.	88.9%
Your parents or your partner’s parents or grandparents	81.7%	75.0%	78.2%	100.0%	73.7%
Other relatives	78.8%	50.0%	58.3%	100.0%	50.0%

**Table 13 healthcare-13-01961-t013:** Frequency and intensity of care in terms of hours spent providing care in the different clusters.

In General, How Many Hours Are You Involved in the Care of the People Listed Above by Providing Personal Care Tasks or Helping with Practical Tasks And/or Their Education?	CL1 N86	CL2 N36	CL3 N99	CL4 N18	CL5 N36
Your partner	3.47	1.52	3.70	6.00	2.32
Your children, younger than 18 years	2.93	1.36	1.15	1.10	1.00
Your children, older than 18 years	4.87	3.56	3.28	1.00	1.56
Your or your partner’s parents or grandparents	2.93	3.01	3.40	2.66	3.10
Other family members	3.03	4.16	3.62	2.86	3.95
Caring for your partner	5.40	75.67	23.23	0	8.64
Caring for your children, younger than 18 yrs	11.26	44.73	55.90	22.18	60.00
Caring for your children, older than 18 yrs	13.80	8.00	8.64	7.00	11.79
Caring for your or your partner’s parents or grandparents	9.99	11.48	4.12	4.36	22.14
Caring for other family members	10.65	32.88	10.53	1.55	22.51

**Table 14 healthcare-13-01961-t014:** Work characteristics.

	CL1 N86	CL2 N36	CL3 N99	CL4 N18	CL5 N36
Hours spent at main job	37.10	43.40	38.80	31.26	38.04
Minutes commuting	45.79	49.12	54.86	24.83	44.44
Possibility of losing one’s job	2.07%	2.00%	1.97%	2.18	2.43
Self-employed	3.6%	9.1%	7.9%	27.8%	8.8%
Public sector	36.1%	28.6%	33.4%	16.7%	29.7%
Making ends meet	3.86	3.75	3.95	3.08	3.23

**Table 15 healthcare-13-01961-t015:** Family and demographic characteristics.

	CL1 N86	CL2 N36	CL3 N99	CL4 N18	CL5 N36
Female	51.8%	37.1%	44.9%	55.6%	44.4%
Social position	5.45	5.65	6.03	4.30	5.17
Political position	5.00	5.05	4.75	4.23	6.20
Health	2.64	2.53	2.78	2.47	2.29
Age	46.31	45.61	39.49	44.98	48.53
Urbanisation (subjective)	3.46	3.01	2.93	2.94	3.04
Attending religious services	5.97	5.55	6.08	7.00	5.91
Household size	3.71	4.03	3.82	3.13	3.65
No. of children living in the household	1.74	2.05	1.77	1.77	1.77
Living with children	73.8%	75.8%	47.3%	88.2%	75.0%
Living with a partner	79.0%	81.8%	60.6%	64.7%	83.3%
Living with grandchildren	5.1%	0.0%	0.0%	0.0%	2.8%
Living with parents or grandparents	41.8%	46.9%	26.9%	5.9%	41.7%
Living with other family members	6.3%	3.1%	22.8%	0.0%	16.2%

**Table 16 healthcare-13-01961-t016:** Working carers and their balancing strategies.

Context	Cluster 1	Cluster 2	Cluster 3	Cluster 4	Cluster 5
Usage of balancing strategies	Minimal (adjusting the care to work)	Medium (using all strategies)	No balancing	Adjusting the care to work (outsourcing the care to formal care)	High balancing (except for outsourcing to formal care)
Usage of structural changes in labour market	Some part-time work and leaving organisation	Some part-time work and leaving organisation	Some changes	Yes, all possible changes (including self-employment)	Yes (including retiring)
Organisational context	Yes	Yes	Supportive	No	Yes
Co-workers	Moderate	Yes	Not reported	Not reported	Yes
Care recipients	Parents Other relatives Adult children	Partners Children under 18	Children under 18 Various other care recipients	People with dementia Children under 18 Parents Other relatives	Children over 18 Parents
Frequency of care	Medium	High	High for children under 18; others low frequency	High	High
Intensity of care	Low	High (partner, child under 18, other relatives)	High for children; low for others	Low	High (children under 18, over 18, parents, other relatives) *
Family structure	Multigenerational household	Multigenerational household	Single family	Single family	Multigenerational household
Demography	Better health Older	Male Higer educated Higher class	Younger Higher class	Female Lower class	Older Poor health Traditional values

Note—* High intensity care amounts to 60 h per week for children under 18 and 12 h for children over 18; 22 h for parents and other relatives.

## Data Availability

The data set is in the process of being submitted to the data archive at the Faculty of Social Sciences. doi:10.17898/10066.

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
