# Peer review of "Work–Care Reconciliation Strategies for a Variety of Informal Carers: What Works and What Does Not?"

_healthcare, 2025, doi:10.3390/healthcare13161961_

Round 1
Reviewer 1 Report
Comments and Suggestions for Authors
I am very pleased to be able to review the manuscript entitled Work-care reconciliation strategies for a variety of informal careers: What works and what doesn't?
This manuscript has the following strengths of the study. In my opinion, the study addresses an issue of enormous social relevance: how careers of family members (whether children, the elderly, partners or chronically ill relatives) manage (or fail) to reconcile these responsibilities with paid work This is an urgent issue, especially in a society that increasingly relies on informal care and at the same time demands labour productivity.
In my opinion, the approach of the study is sound and very interesting. Particularly noteworthy is the fact that careers are not analyzed as a homogeneous group, but that various types of experiences are distinguished according to the link, the health situation of the family member, or the employment context. This differentiation greatly enriches the understanding of the phenomenon.
From a methodological point of view, the use of a nationally representative sample and the analysis of hierarchical clusters to identify profiles of work-life balance strategies provide a robust statistical basis. The use of weighted data is also very successful, which reinforces the validity of the findings.
Finally, the results have a clear applicability: they offer concrete clues for improving public policies to support careers, and can also be useful for companies and organizations committed to work-life balance.
I would like to make some suggestions for improvement and expansion. Although the study is valuable and well developed, there are some aspects that, in my opinion, could be strengthened to broaden its impact:
-Deepen the cultural and comparative context. It would be enriching to broaden the discussion by including comparisons with other European countries that have already implemented innovative work-life balance policies or support for careers. This would allow to better contextualize the findings and open the debate towards possible transferable good practices.
-Clarify the limitations of the design. As a cross-sectional study, the results allow for the detection of associations, but do not allow for the identification of processes or changes over time. It would be advisable to point this out with a little more emphasis, and perhaps suggest future longitudinal studies.
-Review the validity of the instruments. Some of the items used to measure reconciliation strategies seem to have been designed for this study. It would be useful to briefly explain how they were constructed and whether their validity or reliability has been assessed. The possibility of developing and validating more standardized instruments in future research could also be discussed.
-Include a reflection on structural inequalities. Although mentioned in a general way, it would be valuable to go deeper into how socio-economic position, type of employment, or gender affect access to certain work-life balance strategies. This would help to show that not all careers have the same opportunities, and that often those who need it the most receive the least support.
I would like to suggest some literature that could enrich the discussion. In an integrated and natural way, it might be useful to refer to research that has explored complementary dimensions to the focus of the study, such as:
-A systematic review of the role of the informal career and its psychosocial and individual factors, which can provide a deeper framework for the emotional and practical needs of caregiving.
-Studies on caregivers' sense of coherence, resilience, and well-being, which help to explain why some people manage to sustain caregiving without collapsing, while others experience an overwhelming burden.
-Reviews of person-centered care and tools such as the Zarit Burden Scale, which allow for a more accurate assessment of how caregiving affects the subjective level.
-Research looking at specific interventions to reduce the burden of informal caregiving, which could be useful for designing more effective policies.
These contributions would not only complement the discussion but would pave the way for future studies that better integrate structural, emotional, and relational factors of caregiving.
I would like to end my review with a conclusion, as this is a very relevant work, well thought out, and with results that provide clarity on a complex phenomenon. The analysis is rigorous, the conclusions are well supported, and the practical implications are clear.
With minor adjustments (especially in the theoretical framework and in the reflection on inequalities), the article has all that is necessary to be a solid and useful publication for the academic community and policymakers.
Author Response
Review 1
I am very pleased to be able to review the manuscript entitled Work-care reconciliation strategies for a variety of informal careers: What works and what doesn't?
This manuscript has the following strengths of the study. In my opinion, the study addresses an issue of enormous social relevance: how careers of family members (whether children, the elderly, partners or chronically ill relatives) manage (or fail) to reconcile these responsibilities with paid work This is an urgent issue, especially in a society that increasingly relies on informal care and at the same time demands labour productivity. In my opinion, the approach of the study is sound and very interesting. Particularly noteworthy is the fact that careers are not analyzed as a homogeneous group, but that various types of experiences are distinguished according to the link, the health situation of the family member, or the employment context. This differentiation greatly enriches the understanding of the phenomenon. From a methodological point of view, the use of a nationally representative sample and the analysis of hierarchical clusters to identify profiles of work-life balance strategies provide a robust statistical basis. The use of weighted data is also very successful, which reinforces the validity of the findings. Finally, the results have a clear applicability: they offer concrete clues for improving public policies to support careers, and can also be useful for companies and organizations committed to work-life balance. I would like to make some suggestions for improvement and expansion. Although the study is valuable and well developed, there are some aspects that, in my opinion, could be strengthened to broaden its impact:
A: Thank you for taking the time to review the manuscript and for recognising the value of our work. We have carefully and thoroughly reviewed all your comments and made changes that will certainly improve the readability of the article and the presentation of our work. Text was language edited as well.
1) Deepen the cultural and comparative context. It would be enriching to broaden the discussion by inculding comparisons with other European countries that have already implemented innovative work-life balance policies or support for carers. This would allow to better contextualize the findings and open the debate towards possible transferable good practices.
A: We have added comparisons with other European countries in the discussion. Text added:
Lines 701-707: However, the formulation and implementation of policy proposals related to work– care reconciliation and support for working carers requires careful consideration of the specific social, cultural, and institutional contexts. While certain measures can be generalised across the EU (e.g. the Work-Life Balance Directive (EU 2019/1158), which sets a minimum standard of five days' leave for carers and the right to flexible working arrangements), the success and appropriateness of specific measures vary considerably from country to country.
Lines 741-764: In summary, our study shows the different strategies used by working carers in Slovenia to balance their dual role, which are shaped by factors such as the organisational context, care context, household structure and individual characteristics. Our findings suggest that structural changes in labour market participation (e.g. job changes, early retirement or leaving the labour market) are more common among working carers employed in less supportive organisational contexts and among those who primarily care for (older) adults. This illustrates the extent to which reconciliation strategies are influenced not only by the organisational context, but also by the characteristics of those in need of care. Such structural changes were particularly common among carers who provide frequent or intensive care to (older) adults, in contrast to parents of young children, who are more likely to benefit from established reconciliation policies at both national and organisational levels. This shows that one-size-fits-all approach is not sufficient to account for the different realities of working carers. A comparative look at Europe reveals a wide range of supportive measures — such as generous leave policies in the Nordic countries (see [4], [51], [52]), flexible part-time leave options in Germany and Austria and innovative models such as the above-mentioned flexible use of carers' leave in France to reduce working hours - from which Slovenia could draw inspiration. However, for such measures to be truly effective in the Slovenian context, they need to be adapted and responsive to the country’s cultural specificities, care situation, family dynamics, employment relationships and LTC system. Only through such flexibility can the reconciliation of paid work and care become a sustainable and equitable reality for working carers - a reality that supports individual carers, promotes labour market participation and gender equality, and ensures the well-being of both those who provide care and those who receive it.
2) Clarify the limitations of the design. As a cross-sectional study, the results allow for the detection of associations, but do not allow for the identification of processes or changes over time. It would be advisable to point this out with a little more emphasis, and perhaps suggest future longitudinal studies.
A: A new chapter was added after 4th Discussion (i.e. 5th Limitations), highlighting all limitations of the study and suggesting future directions of work. The added text:
Lines: 771-805: One of the main limitations of this study is the use of a cross-sectional survey design, which means we can only hypothesise about changes over time. A longitudinal study would certainly be a way to monitor and understand changes over time by selecting new or already used work–care reconciliation measures.
Since we employed individual items and did not construct scales, no further statistical validation was carried out, only the frequency distributions of the reconciliation strategies were checked and attention was paid to missing values. It makes sense to continue developing the measurement instrument and its validation as part of a more structural statistical approach s a suggestion for further work and for future use in other national contexts.
Clearly one limitation is the fact the distribution of clusters could be influenced by the specific national context, i.e., a country that is currently undergoing a major transition from the scattered and uncoordinated provision of services to a long-term care law. Many countries take a more structured approach to long-term care and provide better recognition of informal carers. In any case, while these reconciliation strategies might also emerge in other national contexts, they most likely will be used differently.
While our study explored work–care reconciliation strategies among working carers, it did not address their emotional impact or how these strategies are perceived by the carers themselves. This represents a significant direction for future research, particularly in examining how such strategies relate to perceived caregiver burden and indicators like the work–life reconciliation index. Recent research shows that informal caregiving often imposes substantial physical and psychological strain; however, protective factors—such as resilience, high self-esteem, and a strong sense of coherence—play a crucial role in maintaining caregiver well-being and adaptability to changing care demands [53,54]. Although findings on intervention effectiveness remain mixed, multicomponent and tailored approaches appear most promising for mitigating caregiver burden [55]. Validated tools such as the Zarit Burden Scale or other tools (see for example [56]) could be employed in future studies to assess how different reconciliation strategies affect subjective burden. Furthermore, there is a growing consensus that interventions for carers should go beyond alleviating caregiver burden and actively promote carers’ resilience and wellbeing by incorporating more holistic frameworks [57] and recognising the diversity of caring contexts - an aspect that is often overlooked [55]. To build on existing knowledge, future longitudinal research is needed to investigate how reconciliation strategies evolve and influence carers’ wellbeing and work–life balance over time.
3) Review the validity of the instruments. Some of the items used to measure reconciliation strategies seem to have been designed for this study. It would be useful to briefly explain how they were constructed and whether their validity or reliability has been assessed. The possibility of developing and validating more standardized instruments in future research could also be discussed.
A: The items used in this study are a mixture of newly developed items and items that have already been used in cross-national surveys such as the European Quality of Life Study (items on flexible working hours). The items developed for this study were derived as follows: A PhD student (Tjaša Potočnik - the co-author) conducted a large qualitative study to identify and classify work-care balance measures used by informal carers of older people. Her work consisted of conducting a large number of qualitative in-depth interviews, followed by a thematic analysis of reconciliation strategies, and she compiled an extensive list of reconciliation strategies. Her dissertation is in the final stages of editing (it has not yet been defended and published). Tjaša was also a young carer who cared for her grandmother during her school years. Co-author Valentina Hlebec is currently leading a basic national project to investigate discrimination against carers of children and older people in the workplace. We have conducted a large number of in-depth interviews and the material is currently in the thematic coding phase. Valentina is a carer of a child with autism and thus brings her life experience of balancing work and care at a public workplace in Slovenia (university). The classification of reconciliation strategies discovered by Tjaša Potočnik was further developed into survey items by Valentina Hlebec. The reconciliation measures were reviewed based on the interview results of the basic project and lived experience and further categorised into structural changes (I have a part-time job. I left my previous job and found a more suitable position in my work organisation. I left my work organisation and found a more suitable position in another work organisation. I left the labour market and found a job as a carer for a family member. I left the labour market and was not employed during the provision of care services. I left my work organisation and became self-employed. I left my job and retired), which may occur once in a caring career, and other more common reconciliation strategies, which may occur daily, were rated according to the frequency of occurrence in the last 12 months (I have taken advantage of the right to a flexible start and end to the working day due to caring responsibilities. I have worked from home to provide care. I have worked overtime due to caring responsibilities to utilise the additional hours for caring. I have used my right to special leave to provide care. I have worked late evenings, nights or early mornings to provide care. I have worked at weekends and/or on public holidays or on holiday to provide care. I have asked my colleagues to take over some of my work duties because I have caring responsibilities. I have given up certain work duties because of caring responsibilities. I have turned down a promotion because of caring responsibilities. I have used some of my annual leave for caring responsibilities. I have used my right to sick leave to take on caring responsibilities because I have caring responsibilities. I have adapted the caring duties to my work schedule. I have hired official carers so that I can concentrate on my work. I asked other family members to do the caring so I could focus on my work. I have taken over caring during my lunch break (e.g. made a phone call or delivered lunch). Before starting the questionnaire, another colleague T.R., who had cared for the mother with dementia in the past, also confirmed that the items were useful.
A longer paragraph was added to the section on methodology. The added text:
Lines 247-271: The items used in this study are a mixture of newly developed items and items that have already been used in cross-national surveys such as the European Quality of Life Study (items on flexible working hours). The items developed for this study were derived as follows: A PhD student (T.P. co-author) conducted a large qualitative study to identify and classify work-care balance measures used by informal carers of older people. Her work consisted of conducting a large number of qualitative in-depth interviews, followed by a thematic analysis of reconciliation strategies, and she compiled an extensive list of reconciliation strategies. Her dissertation is forthcoming. Tjaša was also a young carer who cared for her grandmother during her school years. Co-author V.H is currently leading a basic national project to investigate discrimination against carers of children and/or of older people in the workplace. A large number of in-depth interviews was conducted and the material is currently in the thematic coding phase. V.H is a carer of a child with autism and thus brings her life experience of balancing work and care at a public workplace in Slovenia. The classification of reconciliation strategies were further developed into survey items by V.H. who has extensive teaching and research experience in survey design. The reconciliation measures were reviewed based on the interview results of the basic project and lived experience and further categorised into structural changes which may occur only once in a caring career, and other more common reconciliation strategies, which may occur daily, were rated according to the frequency of occurrence in the last 12 months. Before designing the questionnaire, another colleague T.R., who had cared for the mother with dementia in the past, also confirmed that the items were true to her experience.
A pharagraph was added as a limitation and note for future research. The added text:
Lines 776-781: Since we employed individual items and did not construct scales, no further statistical validation was carried out, only the frequency distributions of the reconciliation strategies were checked and attention was paid to missing values. It makes sense to continue developing the measurement instrument and its validation as part of a more structural statistical approach s a suggestion for further work and for future use in other national contexts.
4) Include a reflection on structural inequalities. Although mentioned in a general way, it would be valuable to go deeper into how socio-economic position, type of employment, or gender affect access to certain work-life balance strategies. This would help to show that not all careers have the same opportunities, and that often those who need it the most receive the least support.
A: In this study some of the socio-economic conditions are already explored, namely: gender, coping with family income, being self employed and such. To make these more visible, we have added subtitles to portions of analysis.
5) I would like to suggest some literature that could enrich the discussion. In an integrated and natural way, it might be useful to refer to research that has explored complementary dimensions to the focus of the study, such as:
-A systematic review of the role of the informal career and its psychosocial and individual factors, which can provide a deeper framework for the emotional and practical needs of caregiving.
-Studies on caregivers' sense of coherence, resilience, and well-being, which help to explain why some people manage to sustain caregiving without collapsing, while others experience an overwhelming burden.
-Reviews of person-centered care and tools such as the Zarit Burden Scale, which allow for a more accurate assessment of how caregiving affects the subjective level.
-Research looking at specific interventions to reduce the burden of informal caregiving, which could be useful for designing more effective policies.
These contributions would not only complement the discussion but would pave the way for future studies that better integrate structural, emotional, and relational factors of caregiving.
A: As our study did not focus on the emotional consequences of different strategies and subjective perceptions of strategies, which are for sure an important aspect to adress in future work. Therefore a paragraph was added as a note for further research:
Lines 788-805: While our study explored work–care reconciliation strategies among working carers, it did not address their emotional impact or how these strategies are perceived by the carers themselves. This represents a significant direction for future research, particularly in examining how such strategies relate to perceived caregiver burden and indicators like the work–life reconciliation index. Recent research shows that informal caregiving often imposes substantial physical and psychological strain; however, protective factors—such as resilience, high self-esteem, and a strong sense of coherence—play a crucial role in maintaining caregiver well-being and adaptability to changing care demands [53,54]. Although findings on intervention effectiveness remain mixed, multicomponent and tailored approaches appear most promising for mitigating caregiver burden [55]. Validated tools such as the Zarit Burden Scale or other tools (see for example [56]) could be employed in future studies to assess how different reconciliation strategies affect subjective burden. Furthermore, there is a growing consensus that interventions for carers should go beyond alleviating caregiver burden and actively promote carers’ resilience and wellbeing by incorporating more holistic frameworks [57] and recognising the diversity of caring contexts - an aspect that is often overlooked [55]. To build on existing knowledge, future longitudinal research is needed to investigate how reconciliation strategies evolve and influence carers’ wellbeing and work–life balance over time.
-------
I would like to end my review with a conclusion, as this is a very relevant work, well thought out, and with results that provide clarity on a complex phenomenon. The analysis is rigorous, the conclusions are well supported, and the practical implications are clear.
With minor adjustments (especially in the theoretical framework and in the reflection on inequalities), the article has all that is necessary to be a solid and useful publication for the academic community and policymakers.
Reviewer 2 Report
Comments and Suggestions for Authors
This study explores the different strategies used by working carers -- individuals who are informal carers and also employees -- to negotiate their informal care responsibilities and work related responsibilities. This study is of great interest and originality in a context, worldwide, where many nations are struggling to respond to all care needs, and healthcare systems are under important pressure. In this context, many nations opt to rely on informal carers to care for those in need. Yet as brought to light in the introduction and literature review of this study, such strategies can have quite dire impact on informal carers and care recipients, at the cost of the health and wellbeing of all parties. Other nations, however, are exploring other strategies, laws, modalities, to support more adequately working carers, notably through various accommodations through employers or benefits.
This study is conducted in the Slovenian context, which, based on the presentation of the national context by the authors, provides a highly relevant case to understand how other nations could explore the reliance on informal carers for the support and needs of its populations. This paper offers an original methodological design, through hierarchical cluster analysis, to create "clusters", acting as sub-groups/profiles, whereby one can better understand the various strategies used by different working carers to respond to their competing responsibilities.
This study, to me, appears well conducted, documented; appropriated and sufficiently described. There are, however, a number of things that need to be revised to make this piece adequate for publication:
- to begin, it is mentioned that this is a "qualitative study" at different places in the paper. This is not the case; it is a quantitative study. Please correct.
- further details need to be provided re: the survey itself. Specifically, the authors should provide an annex with the full questionnaire; or provide information as to (2.1) the number of questions; (2.2) lengths in pages and average time to complete; (2.3) with regard to participation: were there people who did not complete the survey? (2.4) were there any open questions? could the participants input other information than what was provided? (2.5) were there any scale or other questionnaires, validated, that were used to design this survey? (2.6) how was this survey designed? (2.7) what are the competences and expertises of the team to do survey and statistical analysis?
- I would also recommend that the authors adjust sightly their introduction given that, in the end, their participants and context of their data collection, focussed exclusively on working carers who care for a person living with dementia. The needs -- and therefore, work/care related strategies -- available and chosen by informal carers can be influenced by their caring context. These differences are important. I would therefore suggest to the authors that (3.1) they provide a clarification, somewhere in the introduction, that their data collection focussed solely on the context of informal care and caring for a person with dementia, and explain why; and then (3.2) explain how (if) these findings can be used in other contexts of informal carers. This, actually, could be done in the discussion section.
- Discussion. Indeed, I like the discussion, of the results, provided. However, the discussion section should engage more strongly with the current literature (see above for one suggestion), and justify/explore the relevances, applications, for these findings, to other nations, contexts of informal carers, etc.
- Provide a brief conclusion
I am available to read a revised copy of the study.

Author Response
Review 2:
This study explores the different strategies used by working carers -- individuals who are informal carers and also employees -- to negotiate their informal care responsibilities and work related responsibilities. This study is of great interest and originality in a context, worldwide, where many nations are struggling to respond to all care needs, and healthcare systems are under important pressure. In this context, many nations opt to rely on informal carers to care for those in need. Yet as brought to light in the introduction and literature review of this study, such strategies can have quite dire impact on informal carers and care recipients, at the cost of the health and wellbeing of all parties. Other nations, however, are exploring other strategies, laws, modalities, to support more adequately working carers, notably through various accommodations through employers or benefits.
This study is conducted in the Slovenian context, which, based on the presentation of the national context by the authors, provides a highly relevant case to understand how other nations could explore the reliance on informal carers for the support and needs of its populations. This paper offers an original methodological design, through hierarchical cluster analysis, to create "clusters", acting as sub-groups/profiles, whereby one can better understand the various strategies used by different working carers to respond to their competing responsibilities.
This study, to me, appears well conducted, documented; appropriated and sufficiently described. There are, however, a number of things that need to be revised to make this piece adequate for publication:
Thank you for taking the time to review the manuscript and for recognising the value of our work. We have carefully and thoroughly reviewed all your comments and made changes that will certainly improve the readability of the article and the presentation of our work. Text was language edited as well.
Introduction
- Please provide age range
A: Age range provided, text added:
Line 39: (50–65 years of age).
- Unclear; please revise; clarify what you mean for men.
A: Text was revised in order to provide more clarity:
Lines 49-52: Gender inequality is less pronounced among those caring for a partner, as male carers are more likely to take on informal caring tasks in partner relationships than in other caring contexts (e.g. caring for parents, parents-in-law, other relatives)
- Would be of interest to provide the names of such countries, in text.
A: Names of such countries were added in text.
Lines 65-69: In countries where formal care services are well developed and accessible (e.g., Nordic countries like Sweden, Finland and Denmark), research shows the intensity of informal care and issues with reconciling work and informal care are not as acute as in countries where formal care services are unavailable/underdeveloped (such as southern European countries, e.g., Spain, Italy, Portugal).
- Again, would be of interest to share the contexts and countries.
A: We added text, providing context and countries.
Lines 78-80: /…/ affordable and high-quality formal care services and supportive labour policies (generous leave, flexible work options, and protection, e.g., Denmark, Germany, Finland, Sweden, France, Belgium, Austria, UK, Ireland, Lithuania).
Contextual framework
- Why is this the case?
A: Text was added to provide explanation.
Lines 122-124: Care for children is defamilialised due to Slovenia’s generous parental leave, the robust family support policies, and the extensive public childcare system [40–42].
Additionally, we revised a sentence in line XY to provide more accurate description of kinship responsibilities related to care. Text was added:
Lines 126-127: legally obliged to contribute to the cost of long-term care (LTC) services if a care receiver’s income is insufficient [39,43].
- Do you mean 13% of this 15% OR of this 15%, only 2% are non-workers, all the rest of workers? Should be made clearer to avoid questions/confusion
A: The sentences were revised, please see the revised sentence below:
Lines 144-147: According to the EC, LSE and Zigante [5] (pp. 20–21), around 15% of the total population in Slovenia are informal carers, whereas among working age (18–64 years) people 13,1% provide informal care. Among the latter figure, two-thirds (8.3%) are employed and just over one-third are unemployed (4.8%).
Methods
- to begin, it is mentioned that this is a "qualitative study" at different places in the paper. This is not the case; it is a quantitative study. Please correct.
A: Although the hierarchical cluster analysis is exploratory, it is true that this is a quantitative study. In the search for “qualitative” in relation to this particular study has not showed any results. There were some citations of previous work conducted by T.P, which was qualitative.
- further details need to be provided re: the survey itself. Specifically, the authors should provide an annex with the full questionnaire; or provide information as to (2.1) the number of questions; (2.2) lengths in pages and average time to complete; (2.3) with regard to participation: were there people who did not complete the survey? (2.4) were there any open questions? could the participants input other information than what was provided? (2.5) were there any scale or other questionnaires, validated, that were used to design this survey? (2.6) how was this survey designed? (2.7) what are the competences and expertises of the team to do survey and statistical analysis.
A: (2.1) the number of questions: As already stated: A total of 166 variables, including demographic variables, were measured.
(2.2) lengths in pages and average time to complete;
As this was an online survey and respondents could have used a number of different devices, the actual number of pages would differ. We therefore cannot say what was the number of pages a specific respondent has filled in. However: The median survey duration was 14 minutes and 14 seconds.
The added text: The median survey duration was 14 minutes and 14 seconds (Line 217).
(2.3) with regard to participation: were there people who did not complete the survey?
As explained in the methods section: Of the panellists, 1,942 were invited to participate and a total of 1,549 surveys were fully completed, 28 were partially completed. The response rate in relation to the total number of panellists is 15%, while the response rate in relation to the invited respondents is 81%.
(2.4) were there any open questions? could the participants input other information than what was provided?
There were no open-ended questions. Participants had no possibility to add items in this portion of the survey.
2.5) were there any scale or other questionnaires, validated, that were used to design this survey?
The survey questionnaire was longer than the number of items used in this study. All items that are used in this manuscript are reported verbatim (exactly as stated in the survey questionnaire).
A pharagraph was added as a limitation and note for future research. The added text:
Lines 776-781: Since we employed individual items and did not construct scales, no further statistical validation was carried out, only the frequency distributions of the reconciliation strategies were checked and attention was paid to missing values. It makes sense to continue developing the measurement instrument and its validation as part of a more structural statistical approach s a suggestion for further work and for future use in other national contexts.
(2.6) how was this survey designed?
The items used in this study are a mixture of newly developed items and items that have already been used in cross-national surveys such as the European Quality of Life Study (items on flexible working hours). The items developed for this study were derived as follows: A PhD student (Tjaša Potočnik - the co-author) conducted a large qualitative study to identify and classify work-care balance measures used by informal carers of older people. Her work consisted of conducting a large number of qualitative in-depth interviews, followed by a thematic analysis of reconciliation strategies, and she compiled an extensive list of reconciliation strategies. Her dissertation is in the final stages of editing (it has not yet been defended and published). Tjaša was also a young carer who cared for her grandmother during her school years. Co-author Valentina Hlebec is currently leading a basic national project to investigate discrimination against carers of children and older people in the workplace. We have conducted a large number of in-depth interviews and the material is currently in the thematic coding phase. Valentina is a carer of a child with autism and thus brings her life experience of balancing work and care at a public workplace in Slovenia (university). The classification of reconciliation strategies discovered by Tjaša Potočnik was further developed into survey items by Valentina Hlebec. The reconciliation measures were reviewed based on the interview results of the basic project and lived experience and further categorised into structural changes (I have a part-time job. I left my previous job and found a more suitable position in my work organisation. I left my work organisation and found a more suitable position in another work organisation. I left the labour market and found a job as a carer for a family member. I left the labour market and was not employed during the provision of care services. I left my work organisation and became self-employed. I left my job and retired), which may occur once in a caring career, and other more common reconciliation strategies, which may occur daily, were rated according to the frequency of occurrence in the last 12 months (I have taken advantage of the right to a flexible start and end to the working day due to caring responsibilities. I have worked from home to provide care. I have worked overtime due to caring responsibilities to utilise the additional hours for caring. I have used my right to special leave to provide care. I have worked late evenings, nights or early mornings to provide care. I have worked at weekends and/or on public holidays or on holiday to provide care. I have asked my colleagues to take over some of my work duties because I have caring responsibilities. I have given up certain work duties because of caring responsibilities. I have turned down a promotion because of caring responsibilities. I have used some of my annual leave for caring responsibilities. I have used my right to sick leave to take on caring responsibilities because I have caring responsibilities. I have adapted the caring duties to my work schedule. I have hired official carers so that I can concentrate on my work. I asked other family members to do the caring so I could focus on my work. I have taken over caring during my lunch break (e.g. made a phone call or delivered lunch). Before starting the questionnaire, another colleague T.R., who had cared for the mother with dementia in the past, also confirmed that the items were useful.
A longer paragraph was added to the section on methodology. The added text:
Lines 247-271: The items used in this study are a combination of newly developed items and items been used in cross-national surveys such as the European Quality of Life Study (items on flexible working hours). The items developed for this study were derived as follows: A PhD student (T.P., a co-author) conducted a large qualitative study to identify and classify the work–care balance measures used by informal carers of older people. Her work entailed conducting a large number of qualitative in-depth interviews, performing thematic analysis of reconciliation strategies, and compiling an extensive list of reconciliation strategies. Her dissertation[1] is forthcoming. T.P. was also a young carer who during her school years cared for her grandmother. The other co-author, V.H., is currently leading a basic national project to investigate discrimination against carers of children and/or older people in the workplace. A large number of in-depth interviews has been conducted and the material is presently in the thematic coding phase. V.H. is a carer of a child with autism and thus brings her life experience of balancing work and care in a public workplace in Slovenia. The classification of reconciliation strategies was further developed into survey items by V.H., who has extensive teaching and research experience in survey design. The reconciliation measures were reviewed based on interview results of the basic project and lived experience, and further categorised as structural changes which might only occur once in a caring career, and other more common reconciliation strategies, which might occur daily, were rated according to the frequency of occurrence in the last 12 months. Before designing the questionnaire, another colleague, T.R., who had cared for her mother with dementia in the past, additionally confirmed that the items were truly reflected her experience. While T.P. is a qualitative researcher, V.H. holds extensive knowledge of statistics and methodology in both teaching and research. She specialises in survey design with a focus on cognitive laboratory techniques for pre-testing questionnaires.
A paragraph was added as a limitation and note for future research. The added text:
Lines 776-781: Since we employed individual items and did not construct scales, no further statistical validation was carried out, only the frequency distributions of the reconciliation strategies were checked and attention was paid to missing values. It makes sense to continue developing the measurement instrument and its validation as part of a more structural statistical approach s a suggestion for further work and for future use in other national contexts.
2.7. what are the competences and expertises of the team to do survey and statistical analysis?
A: While T.P. is a qualitative researcher, V.H. has extensive knowledge of statistics and methodology, both in teaching and in research. She specialises in survey design with a focus on cognitive laboratory techniques for pre-testing questionnaires. Her sampled methodological publications are:
CASU, Giulia, HLEBEC, Valentina, BOLKO, Irena, et al. Promoting mental health and well-being among adolescent young carers in Europe : a randomized controlled trial protocol. International journal of environmental research and public health. [Online ed.]. 2021, vol. 18, no. 4, 23 str., ilustr. ISSN 1660-4601. DOI: 10.3390/ijerph18042045.
KOPAČ, Gizela, HLEBEC, Valentina. Quality guidelines for mixed methods research in intervention studies : a conceptual model. Advances in methodology and statistics. [Tiskana izd.]. 2020, vol. 17, no. 2, str. 1-29,
TUR-SINAI, Aviad, HLEBEC, Valentina, et al. How many older informal caregivers are there in Europe? : comparison of estimates of their prevalence from three European surveys. International journal of environmental research and public health. [Online ed.]. 2020, vol. 17, no. 24, 17 str., ilustr. ISSN 1660-4601. DOI: 10.3390/ijerph17249531.
KOGOVŠEK, Tina, HLEBEC, Valentina. Measuring personal networks with surveys. Advances in methodology and statistics. [Tiskana izd.]. 2019, vol. 16, no. 2, str. 41-55. ISSN 1854-0023. http://ibmi.mf.uni-lj.si/mz/2019/no-2/Kogovsek2019.pdf, http://ibmi.mf.uni-lj.si/mz/2019/no-2/Kogovsek2019-Supplementary-1.pdf, http://ibmi.mf.uni-lj.si/mz/2019/no-2/Kogovsek2019-Supplementary-2.pdf, http://ibmi.mf.uni-lj.si/mz/2019/no-2/Kogovsek2019-Supplementary-3.pdf, http://ibmi.mf.uni-lj.si/mz/2019/no-2/Kogovsek2019-Supplementary-4.pdf.
MOHORKO, Anja, HLEBEC, Valentina. Degree of cognitive interviewer involvement in questionnaire pretesting on trending survey modes. Computers in human behavior. [Print ed.]. Sep. 2016, vol. 62, str. 79-89, ilustr. ISSN 0747-5632. http://www.sciencedirect.com/science/article/pii/S0747563216301820, DOI: 10.1016/j.chb.2016.03.021.
DYKSTRA, Pearl Annette, PETRIČ, Gregor, PLATINOVŠEK, Rok, KOGOVŠEK, Tina, HLEBEC, Valentina, et al. Social network indices in the Generations and Gender Survey : an appraisal. Demographic research. [Online ed.]. 2016, vol. 34, art. 35, str. 995-1036, ilustr. ISSN 2363-7064. http://www.demographic-research.org/volumes/vol34/35/34-35.pdf, DOI: 10.4054/DemRes.2016.34.35.
MOHORKO, Anja, HLEBEC, Valentina. Effect of a first-time interviewer on cognitive interview quality. Quality & quantity. 2015, vol. 49, no. 5, str. 1897-1918, tabele. ISSN 0033-5177. http://link.springer.com/article/10.1007/s11135-014-0081-0, DOI: 10.1007/s11135-014-0081-0.
HLEBEC, Valentina, KOGOVŠEK, Tina. Different approaches to measure ego-centered social support networks : a meta-analysis. Quality & quantity. 2013, vol. 47, no. 6, str. 3435-3455, tabele. ISSN 0033-5177. DOI: 10.1007/s11135-012-9731-2.
HLEBEC, Valentina, MRZEL, Maja, KOGOVŠEK, Tina. Assessing social support networks in cross-national comparative surveys : measurement issues. Quality & quantity. 2012, vol. 46, no. 5, str. 1431-1449, tabele. ISSN 0033-5177. DOI: 10.1007/s11135-011-9456-7.
And many others.
This has been added in methodological section of the manuscript.
The added text:
Lines 269-271: While T.P. is a qualitative researcher, V.H. holds extensive knowledge of statistics and methodology in both teaching and research. She specialises in survey design with a focus on cognitive laboratory techniques for pre-testing questionnaires.
- I would also recommend that the authors adjust sightly their introduction given that, in the end, their participants and context of their data collection, focussed exclusively on working carers who care for a person living with dementia. The needs -- and therefore, work/care related strategies -- available and chosen by informal carers can be influenced by their caring context. These differences are important. I would therefore suggest to the authors that (3.1) they provide a clarification, somewhere in the introduction, that their data collection focussed solely on the context of informal care and caring for a person with dementia, and explain why; and then (3.2) explain how (if) these findings can be used in other contexts of informal carers. This, actually, could be done in the discussion section.
A: As stated in the methods section: For the part of the analysis presented here, only respondents with a family member in need of care due to dementia or other chronic (long-term) physical or mental illness or disability were considered (i.e. work-life balance strategies). All family carers were included, including those who have recently retired due to caring responsibilities, although most respondents are informal carers.
This is a survey including dementia carers and other type of carers as well.
- Indeed, I like the discussion of the results, provided. However, the discussion section should engage more strongly with the current literature (See above for one suggestion), and justify/explore the relevances, applications, for these findings, to other nations, contexts of informal carers, etc.
A: Discussion section was more strongly engaged with current literature and the following text has been added:
Lines 701-712: However, the formulation and implementation of policy proposals related to work– care reconciliation and support for working carers requires careful consideration of the specific social, cultural, and institutional contexts. While certain measures can be generalised across the EU (e.g. the Work-Life Balance Directive (EU 2019/1158), which sets a minimum standard of five days' leave for carers and the right to flexible working arrangements), the success and appropriateness of specific measures vary considerably from country to country. Nevertheless, recent policy developments in Slovenia, in particular the implementation of the Work-Life Balance Directive (EU 2019/1158) into the Employment Relations Act and the adoption of the Long-Term Care Act in 2023, hold significant potential to reduce existing gaps in work–care reconciliation policies by improving the recognition and support of working informal carers - not only at the individual and organisational level, but also through coordinated support at the national level.
Lines 731-764: Here, the French model of converting leave into reduced working hours or the use of psychosocial support and respite services, as is common in Scandinavian systems, could serve as useful policy templates. Taken together, these cluster-specific findings emphasise the need for a multi-level policy framework that combines universal reconciliation rights with targeted, context-sensitive interventions tailored to the heterogeneity of working carers, their employment and their care situation. Policy measures must be gender-sensitive and sustainable, especially if they aim to promote the sharing of caring responsibilities between women and men.
In summary, our study shows the different strategies used by working carers in Slovenia to balance their dual role, which are shaped by factors such as the organisational context, care context, household structure and individual characteristics. Our findings suggest that structural changes in labour market participation (e.g. job changes, early retirement or leaving the labour market) are more common among working carers employed in less supportive organisational contexts and among those who primarily care for (older) adults. This illustrates the extent to which reconciliation strategies are influenced not only by the organisational context, but also by the characteristics of those in need of care. Such structural changes were particularly common among carers who provide frequent or intensive care to (older) adults, in contrast to parents of young children, who are more likely to benefit from established reconciliation policies at both national and organisational levels. This shows that one-size-fits-all approach is not sufficient to account for the different realities of working carers. A comparative look at Europe reveals a wide range of supportive measures — such as generous leave policies in the Nordic countries (see [4], [51], [52]), flexible part-time leave options in Germany and Austria and innovative models such as the above-mentioned flexible use of carers' leave in France to reduce working hours - from which Slovenia could draw inspiration. However, for such measures to be truly effective in the Slovenian context, they need to be adapted and responsive to the country’s cultural specificities, care situation, family dynamics, employment relationships and LTC system. Only through such flexibility can the reconciliation of paid work and care become a sustainable and equitable reality for working carers - a reality that supports individual carers, promotes labour market participation and gender equality, and ensures the well-being of both those who provide care and those who receive it.
- Provide a brief conclusion
A: We have considered this suggestion, and we have decided against it as we have already extended discussion and provided a new chapter with limitations and future work. We feel this would create repetition in the text.
-------------
I am available to read a revised copy of the study.
Reviewer 3 Report
Comments and Suggestions for Authors
Abstract
- The problem statement is not clear in the abstract
- The study design is not clearly stated
Introduction
- Some long paragraphs need reworking, as it reduce the focus of a reader. i.e paragraph 1-line 26 to 56 , similar to the conceptual framework
- Usually conceptual frameworks are conceptualised in a figure, which is not the case in your study.
Methodology
- Study design not well described; was it a cross-sectional study?
- Did you have any quality controls to enhance data quality? please describe in the section
- How did you go about data management? please describe
Results
- Provide clearer linkage between tables and findings: The narrative could be strengthened to explicitly connect findings across the different tables (Table 9, Table 10, Table 11, Table 12, Table 13). For instance, when discussing structural changes, the text could directly reference how these changes relate to the frequency of reconciliation strategies or the proportion of different types of family carers.
- While statistical significance is indicated, providing effect sizes (e.g., Cramer's V, eta-squared) or confidence intervals for key findings would offer a more complete picture of the magnitude and precision of the observed differences or relationships.
Discussions
- The discussion could include a dedicated section detailing the study's limitations, such as the potential influence of the specific national context (Slovenia) on the findings. This would involve explaining how these limitations might impact the generalizability of the results and suggesting avenues for future research to address them, such as comparative cross-cultural studies
- The discussion should expand on the policy implications derived from the findings, moving beyond general statements; It should offer more specific and actionable recommendations tailored to the different clusters of carers identified in the study, outlining concrete policy interventions that could address their unique challenges and support their reconciliation strategies.
Author Response
Review 3:
Thank you for taking the time to review the manuscript and for recognising the value of our work. We have carefully and thoroughly reviewed all your comments and made changes that will certainly improve the readability of the article and the presentation of our work. Text was language edited as well.
Abstract
- The problem statement is not clear in the abstract
A: The problem statement in the abstract has been reformatted for greater clarity.
- The study design is not clearly stated
A: Text changed in abstract:
Lines 13-16: We conducted a cross-sectional survey to examine structural strategies for work-care balance throughout the caregiving period, followed by the frequency of use of strategies in the last 12 months to better understand what is an effective work–care balance strategy for different working carer types.
Introduction
- Some long paragraphs need reworking, as it reduce the focus of a reader. i.e paragraph 1-line 26 to 56 , similar to the conceptual framework
A: The long paragraph has been split into two parts, improving readability and enhancing the overall clarity of the text.
- Usually conceptual frameworks are conceptualised in a figure, which is not the case in your study.
A: Section “contextual framework” serves primarily to outline the context within which the study was conducted. Since it does not introduce new concepts but rather provides background information, we believe that adding a figure is not necessary in this case.
Methodology
- Study design not well described; was it a cross-sectional study?
A: Text changed in abstract:
Lines 13-16: We conducted a cross-sectional survey to examine structural strategies for work-care balance throughout the caregiving period, followed by the frequency of use of strategies in the last 12 months to better understand what is an effective work–care balance strategy for different working carer types.
- Did you have any quality controls to enhance data quality? please describe in the section
A: Text added in methodological section:
Lines 217-236: The collected units were reviewed for adequacy and data quality according to the following quality criteria.
Basic Quality Criteria:
- Consistency of birth year: register vs. survey response
- Consistency of gender: register vs. survey response
- Household income must be equal to or greater than personal income
- Inconsistencies in employment responses to questions Q20 and Q21
Additional Quality Criteria:
- More than 10 people in the household
- Open-ended questions and numeric input questions
- Has children or has children of a partner and is younger than 20
- Highest personal income class; over 6,000
- Retired and younger than 50
- Holds a PhD and age less than 26
Units where more than one inconsistency was detected based on the above criteria were excluded from the database. A total of 56 units was excluded.
- How did you go about data management? please describe
A: The data is submitted to data archive for consideration. The link is provided at the bottom of the main text before references. The data set is securely stored on the password-protected computer of the principal investigator V.H.
Added text in methodological section:
Lines 236-237: The data set is securely stored on the password-protected computer of principal investigator V.H..
Results
- Provide clearer linkage between tables and findings: The narrative could be strengthened to explicitly connect findings across the different tables (Table 9, Table 10, Table 11, Table 12, Table 13). For instance, when discussing structural changes, the text could directly reference how these changes relate to the frequency of reconciliation strategies or the proportion of different types of family carers.
A: In the revised manuscript, we have strengthened the narrative by explicitly connecting the findings presented in the tables across the Results section, as well as by integrating these links into the Discussion. Text added:
Lines 743-753: Our findings suggest that structural changes in labour market participation (e.g. job changes, early retirement or leaving the labour market) are more common among working carers employed in less supportive organisational contexts and among those who primarily care for (older) adults. This illustrates the extent to which reconciliation strategies are influenced not only by the organisational context, but also by the characteristics of those in need of care. Such structural changes were particularly common among carers who provide frequent or intensive care to (older) adults, in contrast to parents of young children, who are more likely to benefit from established reconciliation policies at both national and organisational levels. This shows that one-size-fits-all approach is not sufficient to account for the different realities of working carers
- While statistical significance is indicated, providing effect sizes (e.g., Cramer's V, eta-squared) or confidence intervals for key findings would offer a more complete picture of the magnitude and precision of the observed differences or relationships.
A: As this is not a study whose purpose is to test hypotheses, but rather to explore the strategies, we do not consider effect sizes or confidence intervals to be very meaningful statistically. The sample size is also not very small, and the sample is statistically representative for adult population. For the Chi2 we provide indication of p value.
Discussions
- The discussion could include a dedicated section detailing the study's limitations, such as the potential influence of the specific national context (Slovenia) on the findings. This would involve explaining how these limitations might impact the generalizability of the results and suggesting avenues for future research to address them, such as comparative cross-cultural studies
- A: Text added in the limitations:
Lines 782-787: Clearly one limitation is the fact the distribution of clusters could be influenced by the specific national context, i.e., a country that is currently undergoing a major transition from the scattered and uncoordinated provision of services to a long-term care law. Many countries take a more structured approach to long-term care and provide better recognition of informal carers. In any case, while these reconciliation strategies might also emerge in other national contexts, they most likely will be used differently.
- The discussion should expand on the policy implications derived from the findings, moving beyond general statements; It should offer more specific and actionable recommendations tailored to the different clusters of carers identified in the study, outlining concrete policy interventions that could address their unique challenges and support their reconciliation strategies.
A: Thank you for the suggestion. We have extended the discussion section to include more specific and actionable policy recommendations tailored to the different clusters of carers identified in the study. Added text in discussion:
Lines 714-739: To ensure these measures are truly effective, they must be adaptable and sensitive to the diverse realities of caregiving. That means recognizing different care arrangements, family structures, and employment situations and giving working carers the flexibility they need to navigate both responsibilities. For example, family carers in Cluster 1, who are likely to be in the early stages of caring, would benefit from early access to flexible working arrangements and awareness-raising campaigns informing them of their rights and entitlements before care demands increase. In contrast, Cluster 4, which consists predominantly of women with low socio-economic status who are employed in less supportive organisational structures, often in the private sector or in precarious forms of employment such as self-employment, points to the need for more robust structural reforms. This could include the introduction of nationally mandated minimum standards for workplace flexibility across all sectors, aimed at reducing the need for carers to make structural adjustments to their labour market participation (e.g. in the form of part-time work or self-employment). Cluster 5 illustrates that even working family carers in a supportive organisational contexts can face negative consequences when caring responsibilities are intensive. For this group, targeted psychosocial services, burnout prevention programmes and the extension of paid care leave would provide much-needed relief and ensure the sustainability of their care arrangements. Here, the French model of converting leave into reduced working hours or the use of psychosocial support and respite services, as is common in Scandinavian systems, could serve as useful policy templates. Taken together, these cluster-specific findings emphasise the need for a multi-level policy framework that combines universal reconciliation rights with targeted, context-sensitive interventions tailored to the heterogeneity of working carers, their employment and their care situation. Policy measures must be gender-sensitive and sustainable, especially if they aim to promote the sharing of caring responsibilities between women and men.
Reviewer 4 Report
Comments and Suggestions for Authors
Dear editor,
Thank you so much for giving me the opportunity to contribute to the Healthcare.
This study focuses on the balance between work and family care among different types of working carers. Based on online survey data from Slovenian adult residents in February 2025, hierarchical clustering analysis was used to identify five distinct caregiver groups. The study provides a detailed analysis of each group's characteristics, their balancing strategies, and specific approaches to achieving work-life equilibrium. We think this topic has some research value. I have just some minor comments.
- Introduction, when discussing gender issues in lines 44-47, the latest research data or trends can be added to enhance the timeliness of the argument.
- Methods, when discussing the weight adjustment method, it is recommended to explain in detail the two weighting methods and the steps and basis of weight adjustment.
- Results, when presenting cluster analysis results, subheadings can be used to clearly distinguish different types of analysis results. In addition, it is recommended to use clear headings and subheadings throughout the article so that readers can better follow the structure of the article.
- Results, when discussing the balance strategies of different caregiver groups, it is suggested to supplement charts or graphics to more intuitively show the frequency and effect of each strategy. In addition, attention should be paid to the adjustment of table format to make it clearer and more beautiful.
- Discussion, when discussing policy proposals, it was suggested that more detailed consideration be given to how these policies could be implemented in different countries or regions, taking into account the differences in social, economic and developmental processes among them.
- Discussion, when discussing the limitations of the study, it is recommended to supplement possible directions for improvement or suggestions for future research.
Author Response
Review 4
Thank you so much for giving me the opportunity to contribute to the Healthcare.
This study focuses on the balance between work and family care among different types of working carers. Based on online survey data from Slovenian adult residents in February 2025, hierarchical clustering analysis was used to identify five distinct caregiver groups. The study provides a detailed analysis of each group's characteristics, their balancing strategies, and specific approaches to achieving work-life equilibrium. We think this topic has some research value. I have just some minor comments.
Thank you for taking the time to review the manuscript and for recognising the value of our work. We have carefully and thoroughly reviewed all your comments and made changes that will certainly improve the readability of the article and the presentation of our work. Text was language edited as well.
- Introduction, when discussing gender issues in lines 44-47, the latest research data or trends can be added to enhance the timeliness of the argument.
A: We have revised the relevant section to strengthen the discussion of gender issues and improve its clarity and relevance. While we did not incorporate new sources, we have added a sentence to provide further context and align the argument more closely with current discourse. We hope this enhancement addresses your concern. Text added:
Lines 46-49: In Europe, the provision of informal care continues to reflect entrenched gender norms, with women disproportionately taking on more complex and time-consuming care tasks. Men are increasingly involved in informal care, but usually with less demanding tasks, reinforcing the primary role of women in informal care provision.
- Methods, when discussing the weight adjustment method, it is recommended to explain in detail the two weighting methods and the steps and basis of weight adjustment.
Text added in methodological section:
Lines 206-212: Two weights were created:
- A weight for all respondents: The weighting variables were caregiver (survey data from June 2024), gender x age, region, type of settlement, and education. The design effect is 2.6, and the increase in sample variance due to the weighting is 160%.
- A weight for caregivers only (based on demographic data of caregivers from the June 2024 survey): The weighting variables were gender x age, region, type of settlement, and education. The design effect is 1.3, and the increase in sample variance due to the weighting is 31%.
- Results, when presenting cluster analysis results, subheadings can be used to clearly distinguish different types of analysis results. In addition, it is recommended to use clear headings and subheadings throughout the article so that readers can better follow the structure of the article.
A: Several subheadings were added throughout results section.
- Results, when discussing the balance strategies of different caregiver groups, it is suggested to supplement charts or graphics to more intuitively show the frequency and effect of each strategy. In addition, attention should be paid to the adjustment of table format to make it clearer and more beautiful.
A: We have considered this comment and we agree with your observation that tables are not beautiful, but the end result needs to be in line with the journal's required table formatting and therefore we cannot freely change it.
- Discussion, when discussing policy proposals, it was suggested that more detailed consideration be given to how these policies could be implemented in different countries or regions, taking into account the differences in social, economic and developmental processes among them.
A: A paragraph was added in the discussion section:
Lines 701-717: However, the formulation and implementation of policy proposals related to work– care reconciliation and support for working carers requires careful consideration of the specific social, cultural, and institutional contexts. While certain measures can be generalised across the EU (e.g. the Work-Life Balance Directive (EU 2019/1158), which sets a minimum standard of five days' leave for carers and the right to flexible working arrangements), the success and appropriateness of specific measures vary considerably from country to country. Nevertheless, recent policy developments in Slovenia, in particular the implementation of the Work-Life Balance Directive (EU 2019/1158) into the Employment Relations Act and the adoption of the Long-Term Care Act in 2023, hold significant potential to reduce existing gaps in work–care reconciliation policies by improving the recognition and support of working informal carers - not only at the individual and organisational level, but also through coordinated support at the national level.
To ensure these measures are truly effective, they must be adaptable and sensitive to the diverse realities of caregiving. That means recognizing different care arrangements, family structures, and employment situations and giving working carers the flexibility they need to navigate both responsibilities.
- Discussion, when discussing the limitations of the study, it is recommended to supplement possible directions for improvement or suggestions for future research.
A: We have added a dedicated paragraph Limitations that outlines the study’s limitations and suggests possible directions for future research.
Text added:
Lines 771-805: 5. Limitations
One of the main limitations of this study is the use of a cross-sectional survey design, which means we can only hypothesise about changes over time. A longitudinal study would certainly be a way to monitor and understand changes over time by selecting new or already used work–care reconciliation measures.
Since we employed individual items and did not construct scales, no further statistical validation was carried out, only the frequency distributions of the reconciliation strategies were checked and attention was paid to missing values. It makes sense to continue developing the measurement instrument and its validation as part of a more structural statistical approach s a suggestion for further work and for future use in other national contexts.
Clearly one limitation is the fact the distribution of clusters could be influenced by the specific national context, i.e., a country that is currently undergoing a major transition from the scattered and uncoordinated provision of services to a long-term care law. Many countries take a more structured approach to long-term care and provide better recognition of informal carers. In any case, while these reconciliation strategies might also emerge in other national contexts, they most likely will be used differently.
While our study explored work–care reconciliation strategies among working carers, it did not address their emotional impact or how these strategies are perceived by the carers themselves. This represents a significant direction for future research, particularly in examining how such strategies relate to perceived caregiver burden and indicators like the work–life reconciliation index. Recent research shows that informal caregiving often imposes substantial physical and psychological strain; however, protective factors—such as resilience, high self-esteem, and a strong sense of coherence—play a crucial role in maintaining caregiver well-being and adaptability to changing care demands [53,54]. Although findings on intervention effectiveness remain mixed, multicomponent and tailored approaches appear most promising for mitigating caregiver burden [55]. Validated tools such as the Zarit Burden Scale or other tools (see for example [56]) could be employed in future studies to assess how different reconciliation strategies affect subjective burden. Furthermore, there is a growing consensus that interventions for carers should go beyond alleviating caregiver burden and actively promote carers’ resilience and wellbeing by incorporating more holistic frameworks [57] and recognising the diversity of caring contexts - an aspect that is often overlooked [55]. To build on existing knowledge, future longitudinal research is needed to investigate how reconciliation strategies evolve and influence carers’ wellbeing and work–life balance over time.
Round 2
Reviewer 2 Report
Comments and Suggestions for Authors
Thank you for this revised copy. The authors have adequately responded to questions, concerns, and revolved the raised issues. This publication is robust, rigorously, and publishable.
Once edits are accepted -- please review carefully the complete to make sure it is free of typos and grammatical errors.